# Contrastive Mutual Information Learning: Toward Robust Representations without Positive-Pair Augmentations

## Abstract

Learning representations that transfer well to diverse downstream tasks remains a central challenge in representation learning. Existing paradigms—contrastive learning, self-supervised masking, and denoising auto-encoders—balance this challenge with different trade-offs. We introduce the *contrastive Mutual Information Machine* (cMIM), a probabilistic framework that extends the Mutual Information Machine (MIM) with a contrastive objective. While MIM maximizes mutual information between inputs and latents and promotes clustering of codes, it falls short on discriminative tasks. cMIM addresses this gap by imposing global discriminative structure while retaining MIM's generative fidelity.

Our contributions are threefold. First, we propose cMIM, a contrastive extension of MIM that removes the need for positive data augmentation and is substantially less sensitive to batch size than InfoNCE. Second, we introduce *informative embeddings*, a general technique for extracting enriched features from encoder–decoder models that boosts discriminative performance without additional training and applies broadly beyond MIM. Third, we provide empirical evidence across vision and molecular benchmarks showing that cMIM outperforms MIM and InfoNCE on classification and regression tasks while preserving competitive reconstruction quality.

These results position cMIM as a unified framework for representation learning, advancing the goal of models that serve both discriminative and generative applications effectively.

## 1 Introduction

Modern representation learning is driven by the promise that a single encoder can produce features that transfer to *unknown* downstream tasks with minimal adaptation. Contrastive methods (*e.g.*, Chen et al. (2020); van den Oord et al. (2018)) have been remarkably successful on this front, but their performance hinges on careful choices of data augmentations to define positives and on large effective numbers of negatives (batch size and/or memory queues). In parallel, generative auto-encoders—including the Mutual Information Machine (MIM) Livne et al. (2019)—optimize likelihood-style objectives and can learn structured latent spaces without augmentation, yet their representations often underperform on discriminative tasks compared to contrastive counterparts. This leaves a practical gap: how can we endow generative models with *global discriminative structure* while avoiding the brittleness of augmentation design and batch-size sensitivity?

**Problem.** We seek a self-supervised framework that (i) learns discriminative features *without explicit positive pairs*, (ii) is *robust* to the number of in-batch negatives, and (iii) *preserves generative fidelity* so that reconstructions and likelihood proxies do not degrade. The solution should apply to encoder–decoder architectures and support simple, post-hoc embedding extraction for downstream tasks.

**Our approach.** We introduce *cMIM* (Contrastive MIM), which integrates a contrastive term into MIM by introducing a binary variable $k$ indicating whether $(x, z)$ is a matched pair. The resulting objective uses an *in-batch expectation* over mismatched $(x, z)$ pairs to produce contrast *without* positive augmentations. Algebraically (Sec. 2), the negative log-probability of $k{=}1$ is equivalent to

an InfoNCE loss where the positive logit is shifted by $\log(B-1)$, yielding distinct calibration and reduced sensitivity to batch size while MIM supplies local attraction. Together, cMIM encourages *angular* separation among dissimilar samples and *radial* clustering for similar samples, improving downstream separability while preserving reconstruction.

**Contributions.**

1. **Contrastive MIM objective.** We extend MIM with a contrastive discriminator over $(x, z)$ that *does not require positive data augmentation* and is empirically *less sensitive to batch size* than InfoNCE. We establish its connection to InfoNCE via a fixed positive-logit offset and provide a concentration bound explaining batch-size robustness.

2. **Informative embeddings.** We propose a generic way to extract *informative embeddings* from encoder–decoder models by reusing decoder hidden states immediately before parameterizing $p_\theta(x \mid z)$. This improves discriminative performance *without* extra training and applies broadly to pre-trained encoder–decoder architectures.

3. **Empirical validation.** Across MNIST-like image classification and molecular property prediction, cMIM matches MIM on reconstruction while achieving higher downstream accuracy/rank on average, and exhibits low batch-size sensitivity in controlled analyses.

By coupling generative modeling with a calibrated contrastive signal, cMIM moves toward a *single*, augmentation-light framework that serves both discriminative and generative use cases.

## 2 FORMULATION

We extend the Mutual Information Machine (MIM)—a probabilistic auto-encoder that maximizes mutual information and promotes clustered latents—with a contrastive objective to add global discriminative structure while preserving generative fidelity. Throughout, $X$ denotes observations and $Z$ latent codes. Our extension, *cMIM*, retains MIM's local Euclidean clustering and adds angular separation between dissimilar samples, improving downstream discrimination without requiring positive data augmentations.

### 2.1 CONTRASTIVE LEARNING

Contrastive learning maximizes similarity of positive pairs and minimizes that of negatives, often with cosine similarity $s\left(\boldsymbol{z}_i, \boldsymbol{z}_j\right) = \frac{\boldsymbol{z}_i \cdot \boldsymbol{z}_j}{\|\boldsymbol{z}_i\| \|\boldsymbol{z}_j\|}$ and temperature-scaled logits $g\left(\boldsymbol{z}_i, \boldsymbol{z}_j\right) \equiv g_{ij} = \exp(s\left(\boldsymbol{z}_i, \boldsymbol{z}_j\right) / \tau)$. The per-sample InfoNCE objective van den Oord et al. (2018) is

$$\text{InfoNCE}(\boldsymbol{x}_i, \boldsymbol{x}_i^+) = -\log\left(\frac{g\left(\boldsymbol{z}_i, \boldsymbol{z}_i^+\right)}{\sum_{j=1}^B g\left(\boldsymbol{z}_i, \boldsymbol{z}_j\right)}\right), \tag{1}$$

with $\boldsymbol{x}_i^+$ a positive augmentation of $\boldsymbol{x}_i$ and $\{\boldsymbol{x}_j\}$ negatives from other sources. In practice, it becomes a $B$-way classification over logits $\{s\left(\boldsymbol{z}_i, \boldsymbol{z}_j\right) / \tau\}_{j=1}^B$ and is sensitive to augmentation design and batch size.

### 2.2 CONTRASTIVE MIM LEARNING (CMIM)

We augment MIM with a binary variable k (see Fig. 7d-e in Appendix B for a graphical model) to induce contrast without data augmentation. The corresponding joint distributions factor as

$$q_{\boldsymbol{\theta}}(\boldsymbol{x}, \boldsymbol{z}, \text{k}) = q_{\boldsymbol{\theta}}(\text{k}|\boldsymbol{x}, \boldsymbol{z}) \, q_{\boldsymbol{\theta}}(\boldsymbol{z}|\boldsymbol{x}) \, q_{\boldsymbol{\theta}}(\boldsymbol{x}), \qquad p_{\boldsymbol{\theta}}(\boldsymbol{x}, \boldsymbol{z}, \text{k}) = p_{\boldsymbol{\theta}}(\text{k}|\boldsymbol{x}, \boldsymbol{z}) \, p_{\boldsymbol{\theta}}(\boldsymbol{x}|\boldsymbol{z}) \, p_{\boldsymbol{\theta}}(\boldsymbol{z}). \tag{2}$$

Let $\boldsymbol{z}_i \sim q_{\boldsymbol{\theta}}(\boldsymbol{z}|\boldsymbol{x}_i)$ be the latent for $\boldsymbol{x}_i$. We set $\text{k} = 1$ for the matched pair $(\boldsymbol{x}_i, \boldsymbol{z}_i)$ and $\text{k} = 0$ for mismatched pairs $(\boldsymbol{x}_i, \boldsymbol{z}_j)$ when $j \neq i$. Using cosine similarity, we define shared encoder/decoder discriminators

$$q_{\boldsymbol{\theta}}(\text{k} \mid \boldsymbol{z} = \boldsymbol{z}_i, \boldsymbol{x}) = p_{\boldsymbol{\theta}}(\text{k} \mid \boldsymbol{z} = \boldsymbol{z}_i, \boldsymbol{x}) = \text{Bernoulli}(\text{k}; p_{k=1}), \tag{3}$$

with

$$p_{k=1}(\boldsymbol{x}_i, \boldsymbol{z}_i) = \frac{g_{ii}}{g_{ii} + \mathbb{E}_{\boldsymbol{x}' \sim \mathcal{P}(\boldsymbol{x}), \boldsymbol{z}' \sim q_{\boldsymbol{\theta}}(\boldsymbol{z}|\boldsymbol{x}')} \left[g\left(\boldsymbol{z}_i, \boldsymbol{z}'\right)\right]} \approx \frac{g_{ii}}{g_{ii} + \frac{1}{B-1} \sum_{\substack{j=1 \\ j \neq i}}^B g_{ij}}. \tag{4}$$

---

**Algorithm 1** Learning parameters $\boldsymbol{\theta}$ of cMIM

---

**Require:** Samples from dataset $\mathcal{P}(\boldsymbol{x})$
1: **while** not converged **do**
2:     $\mathcal{D} \leftarrow \{\boldsymbol{x}_j, \boldsymbol{z}_j \sim q_{\boldsymbol{\theta}}(\boldsymbol{z}|\boldsymbol{x})\mathcal{P}(\boldsymbol{x})\}_{j=1}^B$ {*Sample a batch*}
3:     $\hat{\mathcal{L}}_{\text{A-MIM}}(\boldsymbol{\theta}; \mathcal{D}) = -\frac{1}{B}\sum_{i=1}^B \big( \log p_{\boldsymbol{\theta}}(\boldsymbol{x}_i|\boldsymbol{z}_i) + \log p_{k=1}(\boldsymbol{x}_i, \boldsymbol{z}_i) + \frac{1}{2}(\log q_{\boldsymbol{\theta}}(\boldsymbol{z}_i|\boldsymbol{x}_i) + \log \mathcal{P}(\boldsymbol{z}_i)) \big)$
4:     $\Delta\boldsymbol{\theta} \propto -\nabla_{\boldsymbol{\theta}}\hat{\mathcal{L}}_{\text{A-MIM}}(\boldsymbol{\theta}; \mathcal{D})$ {*Reparameterized gradients*}
5: **end while**

---

where $B$ is the batch size, and the expectation is approximated using all negative examples in the batch. During training we always have k $= 1$; negatives act implicitly through the expectation in $p_{k=1}$, enabling a contrastive signal without explicit positive augmentations. This expectation form reduces sensitivity to batch size with likely error proportional to $\mathcal{O}(1/(B-1))$; see Appendix C for the concentration bound (Eq. (11)) via Hoeffding's inequality Hoeffding (1963).

### 2.2.1 cMIM TRAINING PROCEDURE

Training follows the MIM objective over the extended model Livne et al. (2019) which includes parametrized join probability models

$$\mathcal{M}_{\boldsymbol{\theta}}(\boldsymbol{x}, \boldsymbol{z}, \text{k}) = \frac{1}{2}\left(p_{\boldsymbol{\theta}}(\text{k} \mid \boldsymbol{z}, \boldsymbol{x})\, p_{\boldsymbol{\theta}}(\boldsymbol{x} \mid \boldsymbol{z})\, p_{\boldsymbol{\theta}}(\boldsymbol{z}) + q_{\boldsymbol{\theta}}(\text{k} \mid \boldsymbol{z}, \boldsymbol{x})\, q_{\boldsymbol{\theta}}(\boldsymbol{z} \mid \boldsymbol{x})\, q_{\boldsymbol{\theta}}(\boldsymbol{x})\right), \quad (5)$$

and corresponding sampling distribution

$$\mathcal{M}_{\mathcal{S}}(\boldsymbol{x}, \boldsymbol{z}, \text{k}) = \frac{1}{2}\left(p_{\boldsymbol{\theta}}(\text{k}|\boldsymbol{z}, \boldsymbol{x})\, p_{\boldsymbol{\theta}}(\boldsymbol{x}|\boldsymbol{z})\, \mathcal{P}(\boldsymbol{z}) + q_{\boldsymbol{\theta}}(\text{k}|\boldsymbol{z}, \boldsymbol{x})\, q_{\boldsymbol{\theta}}(\boldsymbol{z}|\boldsymbol{x})\, \mathcal{P}(\boldsymbol{x})\right), \quad (6)$$

where $\mathcal{P}(\boldsymbol{z})$ is a Normal anchor distribution, and $\mathcal{P}(\boldsymbol{x})$ is the data distribution. MIM minimizes the symmetric cross-entropy between $\mathcal{M}_{\boldsymbol{\theta}}$ and $\mathcal{M}_{\mathcal{S}}$, yielding an upper bound

$$\begin{aligned}\mathcal{L}_{\text{MIM}}(\boldsymbol{\theta}) =& \frac{1}{2}\Big(CE\left(\mathcal{M}_{\mathcal{S}}(\boldsymbol{x}, \boldsymbol{z}, \text{k}),\, q_{\boldsymbol{\theta}}(\boldsymbol{x}, \boldsymbol{z}, \text{k})\right) + CE\left(\mathcal{M}_{\mathcal{S}}(\boldsymbol{x}, \boldsymbol{z}, \text{k}),\, p_{\boldsymbol{\theta}}(\boldsymbol{x}, \boldsymbol{z}, \text{k})\right)\Big) \\ \geq&\, H_{\mathcal{M}_{\mathcal{S}}}(\boldsymbol{x}, \text{k}) + H_{\mathcal{M}_{\mathcal{S}}}(\boldsymbol{z}) - I_{\mathcal{M}_{\mathcal{S}}}(\boldsymbol{x}, \text{k}; \boldsymbol{z}),\end{aligned} \quad (7)$$

treating $(\boldsymbol{x}, \text{k})$ as observed (with k $\equiv 1$). The empirical A-MIM loss used in Alg. 1 is

$$\mathcal{L}_{\text{A-MIM}}(\boldsymbol{\theta}) = -\frac{1}{2}\mathbb{E}_{\boldsymbol{x}\sim\mathcal{P}(\boldsymbol{x}),\boldsymbol{z}\sim q_{\boldsymbol{\theta}}(\boldsymbol{z}|\boldsymbol{x}),\text{k}=1}\begin{bmatrix}\log p_{\boldsymbol{\theta}}(\text{k}|\boldsymbol{z}, \boldsymbol{x}) + \log p_{\boldsymbol{\theta}}(\boldsymbol{x}|\boldsymbol{z}) + \log p_{\boldsymbol{\theta}}(\boldsymbol{z})+\\ \log q_{\boldsymbol{\theta}}(\text{k}|\boldsymbol{z}, \boldsymbol{x}) + \log q_{\boldsymbol{\theta}}(\boldsymbol{z}|\boldsymbol{x}) + \log q_{\boldsymbol{\theta}}(\boldsymbol{x})\end{bmatrix} \quad (8)$$

with the final empirical objective

$$\hat{\mathcal{L}}_{\text{A-MIM}}(\boldsymbol{\theta}; \mathcal{D}) = -\frac{1}{N}\sum_{i=1}^N\left(\log p_{\boldsymbol{\theta}}(\boldsymbol{x}_i|\boldsymbol{z}_i) + \log p_{k=1}(\boldsymbol{x}_i, \boldsymbol{z}_i) + \frac{1}{2}(\log q_{\boldsymbol{\theta}}(\boldsymbol{z}_i|\boldsymbol{x}_i) + \log \mathcal{P}(\boldsymbol{z}_i))\right), \quad (9)$$

where $\mathcal{D} = \{\boldsymbol{x}_i, \boldsymbol{z}_i \sim q_{\boldsymbol{\theta}}(\boldsymbol{z}|\boldsymbol{x})\mathcal{P}(\boldsymbol{x})\}_{i=1}^N$, $p_{k=1}$ from Eq. (4) is used symmetrically, $\mathcal{P}(\boldsymbol{z}_i) = \mathcal{N}(\boldsymbol{z}_i; 0, 1)$ anchors the latents; and the model marginal distributions (under the model mixture) are defined as $p_{\boldsymbol{\theta}}(\boldsymbol{z}) = \mathbb{E}_{\boldsymbol{x}}[q_{\boldsymbol{\theta}}(\boldsymbol{z}|\boldsymbol{x})]$, $q_{\boldsymbol{\theta}}(\boldsymbol{x}) = \mathbb{E}_{\boldsymbol{z}\sim\mathcal{P}(\boldsymbol{z})}[p_{\boldsymbol{\theta}}(\boldsymbol{x}|\boldsymbol{z})]$.

### 2.2.2 CONTRASTIVE MIM AND INFONCE

**High-level relation.** Using the algebra in Appendix C, Eq. (12), $-\log p_{k=1}$ is equivalent to an InfoNCE loss computed on logits where the positive is shifted by $\log(B-1)$; i.e., InfoNCE with a fixed positive-logit offset. This yields different calibration (equal logits $\Rightarrow p_{k=1} = 1/2$) and focuses gradients on the negative mean under cosine similarity (the MIM term supplies local attraction). Our mean-denominator form also explains cMIM's reduced sensitivity to batch size, while still benefiting from more negatives (e.g., via memory queues). cMIM retains MIM's mutual-information bound (over $I_{\mathcal{M}_{\mathcal{S}}}(\boldsymbol{x}, \text{k}; \boldsymbol{z})$, equivalent to $I_{\mathcal{M}_{\mathcal{S}}}(\boldsymbol{x}; \boldsymbol{z})$ since k $\equiv 1$), but does not enjoy the classical InfoNCE MI bound.

**Complexity.** The contrastive term uses all in-batch mismatches via a mean over $B-1$ negatives, so its computational and memory costs are $O(B)$ per anchor (matching standard InfoNCE); an optional memory queue of size $M$ trades compute for stability with $O(M)$ similarity evaluations. We do not use memory queues in our experiments; all results are in-batch.

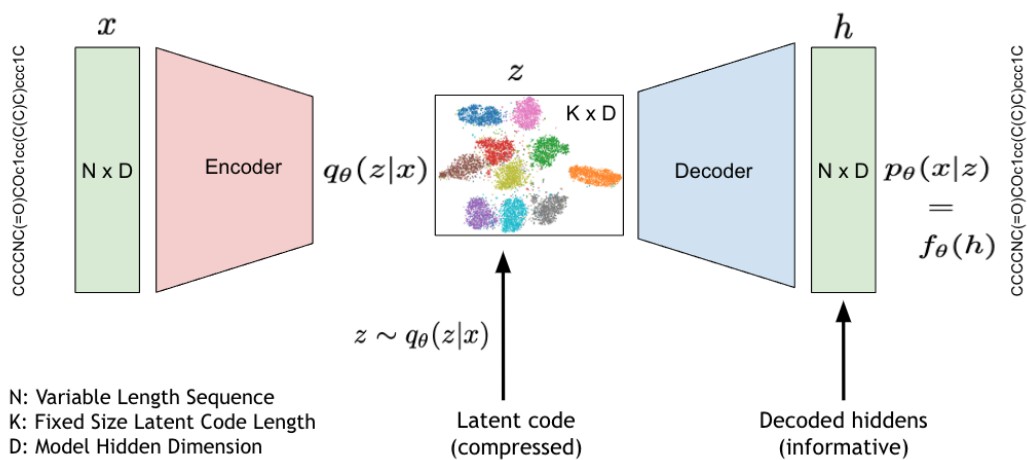

Figure 1: Informative embeddings $h$ are extracted from an input $x$ using the decoder's hidden states prior to projection to $p_\theta$'s parameters. For auto-regressive decoders, use teacher forcing.

## 2.3 INFORMATIVE EMBEDDINGS

As an additional contribution, we propose to extract *informative embeddings* $h$ (depicted in Fig. 1) from the decoder's hidden states immediately before parameterization of $p_\theta(x \mid z) = f_\theta(h)$, then reuse $h$ for downstream discriminative tasks such as classification or regression. For auto-regressive decoders, we employ teacher forcing; for non-autoregressive decoders (e.g., images) $h$ is used directly. Formally,

$$h_i = \mathrm{Decoder}(x_i \mid z_i \sim q_\theta(z \mid x_i)) = \mathrm{Decoder}(x_i, \mathrm{Encoder}(x_i)), \tag{10}$$

optionally mean-pooled over sequence length. This produces enriched features that reflect both the latent code and the decoder's predictive context, and in our experiments improves downstream discriminative performance without additional training. We note that the goal here is to enrich the representations for downstream tasks, and we did not find it to be better in unsupervised clustering.

## 3 EXPERIMENTS

We evaluate cMIM on (i) a controlled 2D toy setting that isolates the effect of the contrastive term, (ii) MNIST-like image datasets for representation quality under downstream classification, and (iii) molecular property prediction on ZINC15 (Sterling & Irwin, 2015). We further study batch-size robustness, reconstruction quality, and ablations.

### 3.1 EXPERIMENT DETAILS AND DATASETS

All models are trained fully unsupervised. Unless noted otherwise, the encoder parameterizes a Gaussian posterior (mean and variance), with the predicted variance clamped to a minimum of *1e-6* for numerical stability. For each run we select the checkpoint with the lowest validation loss; we do *not* monitor downstream accuracy during training and we avoid hand-picking intermediate checkpoints. For downstream evaluation, we freeze the encoder–decoder and train lightweight classifiers on top of learned representations using the held-out test split. This protocol aims to compare the quality of unsupervised representations rather than checkpoint-selection heuristics. Full datasets, architectural and optimization details appear in Appendix D.

**2D Toy Example.** We generate 1,000 points in $\mathbb{R}^2$, initialized in the first quadrant, and examine the effect of the contrastive MIM term in Eq. (4) on the learned latent codes.

**Image Classification on MNIST-like Datasets.** We train MIM, cMIM, VAE, AE, and InfoNCE to convergence on MNIST-like datasets, and compare representations on downstream classification

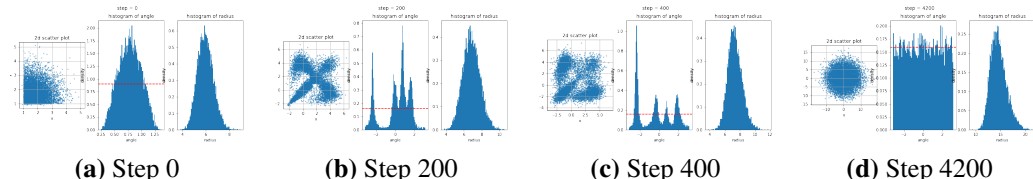

**(a)** Step 0      **(b)** Step 200      **(c)** Step 400      **(d)** Step 4200

Figure 2: Effect of the contrastive term in Eq. (4) on the 2D example. Each panel shows the latent space (left), the histogram of latent angles (middle), and the histogram of latent radii (right). From **(a)** initialization to **(d)** after 4,200 steps, the angles become approximately uniform while radial variability is preserved. This yields angular separation complementary to MIM's radial clustering, improving downstream separability.

tasks while probing sensitivity to batch size. Datasets include MNIST (Deng, 2012), FashionMNIST (Xiao et al., 2017), EMNIST (Cohen et al., 2017), and MedMNIST (Yang et al., 2021); see Table 3 in Appendix D.1. All images are resized to $28 \times 28$ and converted to Black & White if needed. We use $\tau = 0.1$ (van den Oord et al., 2018) following a small hyper-parameter search of $\tau \in \{0.1, 1\}$. The encoder is a Perceiver (Jaegle et al., 2021) with one cross-attention layer and four self-attention layers (hidden size 16), projecting 784 pixels to 400 steps, followed by a projection to 64-dimensional latents; the decoder mirrors this design. This simple architecture induces a strong inductive bias that favors AE without additional regularization (Tschannen et al., 2018). Models are trained for 1M steps with batch sizes $\{2, 5, 10, 100, 200\}$ using Adam $(10^{-3})$ and a WSD scheduler (Hu et al., 2024). Classifiers are KNN ($k$=5; cosine and Euclidean) and a one-hidden-layer MLP (width 400; Adam $10^{-3}$; 1,000 steps). We applied data augmentation as a regularization technique for all models, independent of additional positive samples that are required for InfoNCE. See data augmentation description in Appendix D.1.

**Molecular Property Prediction.** Following Reidenbach et al. (2023), we train on ZINC15 (Sterling & Irwin, 2015) with SMILES (Weininger, 1988). Tasks include regression of ESOL, FreeSolv, and Lipophilicity. Here $\tau = 1$. MIM and cMIM are trained for 250k steps on 723M training molecules (dataset construction, model sizes, tokenizer, and optimization in Appendix D). We evaluate SVM and MLP regressors trained on either mean encodings or informative embeddings (Sec. 2.3), and compare with CDDD (Winter et al., 2019), MegaMolBART (Irwin et al., 2022), Perceiver, VAE, and Morgan fingerprints.

### 3.2 EFFECTS OF CMIM LOSS ON 2D TOY EXAMPLE

We minimize the negative log-likelihood induced by Eq. (4) with $\tau = 1$ in two latent dimensions. As predicted by hyperspherical uniformity analyses (Wang & Isola, 2020), the learned codes spread uniformly in angle while maintaining a non-degenerate radial distribution (Fig. 2). The contrastive term integrates with MIM's local attraction, preserving radial clustering and adding global angular structure.

### 3.3 CLASSIFICATION ACCURACY

We treat classification accuracy as a proxy for representation quality (never used for training or model selection). All models share the same backbone; InfoNCE uses only the encoder. We evaluate checkpoints with the lowest validation loss to control for optimization length, architecture, and data usage.

We report KNN classification accuracy (Cosine and Euclidean) which measures clustering, and a one-hidden-layer MLP classification accuracy which measures the information content of the embeddings. Inputs are only mean encodings here, since InfoNCE does not support informative embeddings (Sec. 2.3). For each model and batch size we evaluate 6 settings (3 classifiers × 2 embedding types) across 15 datasets, yielding 90 tasks (45 for InfoNCE which does not support informative embeddings). We summarize by (i) the average z-normalized accuracy per dataset/evaluation (z-scores computed across all models and batch sizes) and (ii) the average rank (Fig. 3). cMIM achieves top

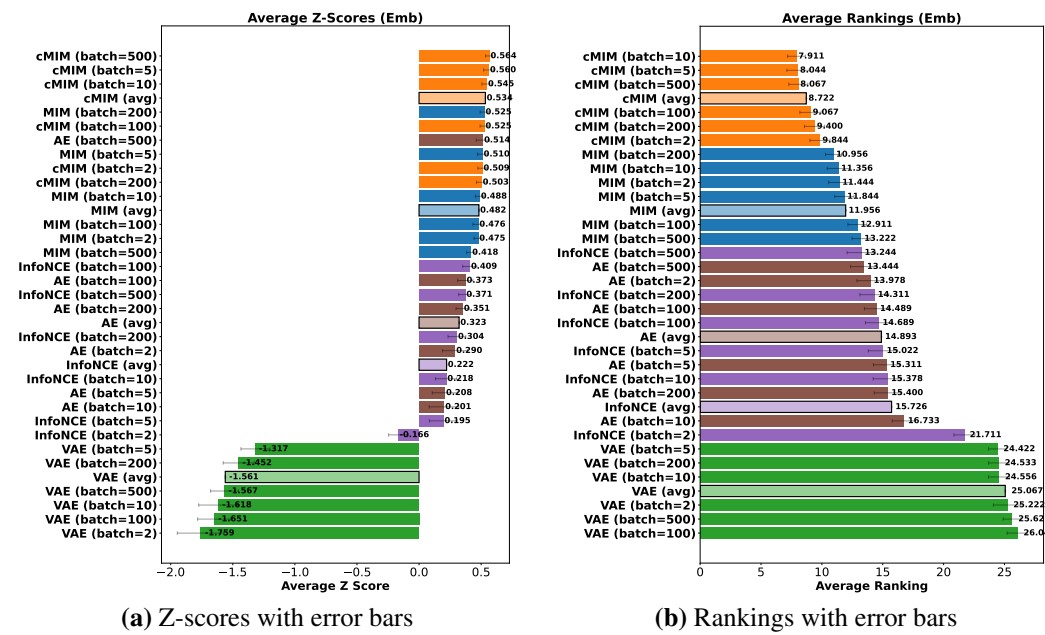

**(a)** Z-scores with error bars    **(b)** Rankings with error bars

Figure 3: Classification accuracy across datasets and classifiers. Only regular embeddings were used here. Colors indicate model families: cMIM (orange), MIM (blue), InfoNCE (purple), VAE (green), AE (brown). Light shades with black frames denote model averages. Across batch sizes and metrics, cMIM attains the best average z-score and ranking.

| Model (Latent K × D) | ESOL | | FreeSolv | | Lipophilicity | | Recon. |
|---|---|---|---|---|---|---|---|
| | SVM | MLP | SVM | MLP | SVM | MLP | |
| MIM (1 × 512) | 0.65 | 0.34 | 2.23 | 1.82 | 0.663 | 0.61 | 100% |
| cMIM (1 × 512) | 0.47 | 0.19 | 2.32 | 1.67 | 0.546 | 0.38 | 100% |
| MIM (1 × 512) info emb | 0.21 | 0.29 | 1.55 | 1.40 | 0.234 | 0.28 | 100% |
| cMIM (1 × 512) info emb | 0.21 | 0.24 | 1.74 | 1.35 | 0.24 | 0.23 | 100% |
| CDDD (512) | **0.33** | – | **0.94** | – | **0.40** | – | – |
| †MegaMolBART (N × 512) | 0.37 | 0.43 | 1.24 | 1.40 | 0.46 | 0.61 | 100% |
| †Perceiver (4 × 512) | 0.40 | 0.36 | 1.22 | 1.05 | 0.48 | 0.47 | 100% |
| †VAE (4 × 512) | 0.55 | 0.49 | 1.65 | 3.30 | 0.63 | 0.55 | 46% |
| Morgan fingerprints (512) | 1.52 | 1.26 | 5.09 | 3.94 | 0.63 | 0.61 | – |

Table 1: Molecular property prediction using model embeddings and informative embeddings (where indicated). Lower RMSE is better for the regression errors reported. Models marked † are from Reidenbach et al. (2023). Bold: best non-MIM result. Highlight: best among MIM-based models. Despite being trained without property supervision, cMIM with informative embeddings is competitive with, and in some cases better than, the baselines.

or near-top performance across batch sizes and classifiers. Additional detailed and complete results can found in Appendix E.2, including informative embeddings results.

**Molecular Property Prediction and Informative Embeddings.** Table 1 compares MIM and cMIM on ESOL, FreeSolv, and Lipophilicity regression tasks using SVM and MLP regressors trained on (i) mean encodings and (ii) informative embeddings. Baselines include CDDD (Winter et al., 2019), MegaMolBART (Irwin et al., 2022), Perceiver, VAE, and Morgan fingerprints,

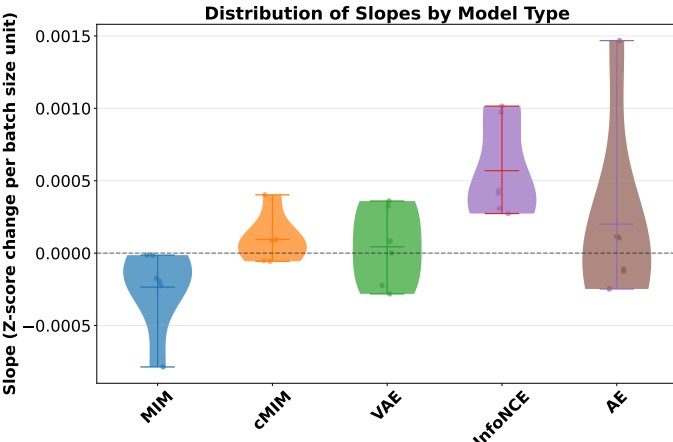

| Model | p-value | n |
|---|---|---|
| **MIM** † | **0.02938** | 90 |
| cMIM | 0.1775 | 90 |
| VAE | 0.787 | 90 |
| **InfoNCE** † | $7.6 \times 10^{-8}$ | 45 |
| AE | 0.1028 | 90 |

Table 2: Two-sided $t$-test of average slope $\neq 0$ (batch-size sensitivity). **Bold** † indicates statistical significance ($p < 0.05$). InfoNCE shows clear dependence on batch size; cMIM does not.

Figure 4: Distribution of slopes from linear fits of accuracy vs. batch size. Each point is the average z-score (over datasets) for a model trained on MNIST-like data under a given evaluation setting. cMIM exhibits the tightest distribution centered near zero, indicating robustness to batch-size variation.

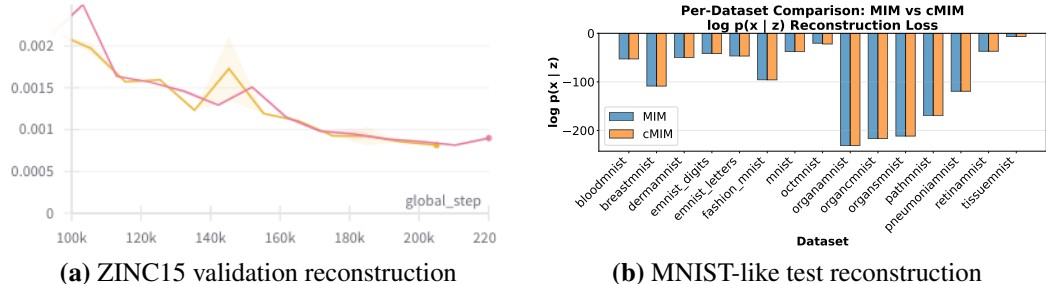

**(a)** ZINC15 validation reconstruction      **(b)** MNIST-like test reconstruction

Figure 5: Reconstruction performance of MIM vs. cMIM. **(a)** Validation reconstruction during molecular training (cMIM yellow, MIM pink) is comparable. **(b)** Per-dataset test reconstruction log-likelihood on MNIST-like data is similarly close. The contrastive term does not degrade reconstruction quality.

following Reidenbach et al. (2023). We note that CDDD is trained with the regression tasks here as a regularization term. cMIM with and without informative embeddings improves over vanilla MIM and is competitive with strong baselines, underscoring the utility of informative embeddings and the global discriminative structure encouraged by cMIM.

### 3.4 BATCH SIZE SENSITIVITY

For each model we regress average z-score (over datasets) against batch size across six evaluation settings (three classifiers × two embedding types). The slope summarizes sensitivity: positive slopes indicate accuracy increases with larger batches, while near-zero slopes indicate robustness. Detailed per-dataset results (90 experiments) appear in Appendix E.1. Figure 4 shows that cMIM has both the smallest spread and mean slope near zero. The statistical test in Table 2 confirms that InfoNCE is batch-size sensitive ($p \ll 0.05$), whereas cMIM is not significant at the same level.

### 3.5 RECONSTRUCTION

Across both molecular data and MNIST-like images, cMIM matches MIM on reconstruction (Fig. 5), which we use as a proxy for generative fidelity in our setup.

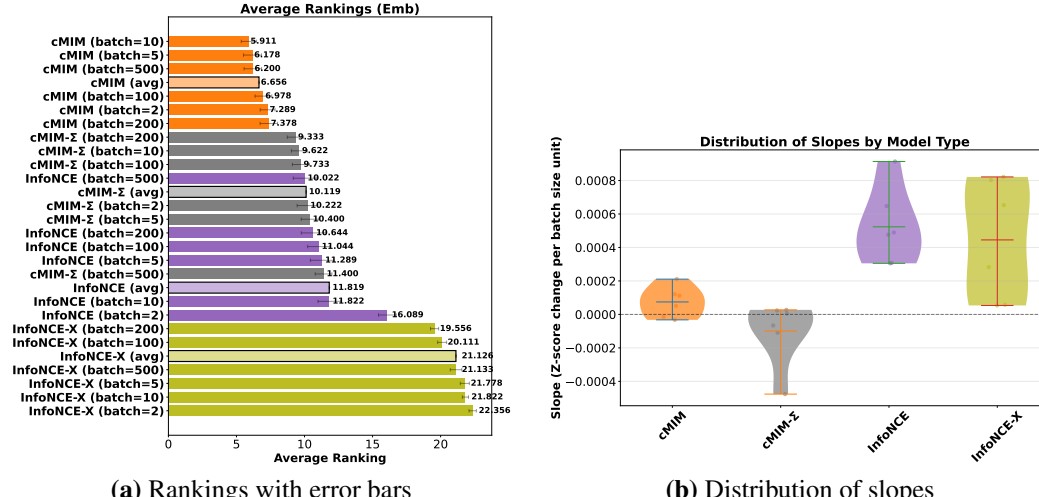

(a) Rankings with error bars    (b) Distribution of slopes

Figure 6: Ablations on MNIST-like data. Only regular embeddings were used here. Colors: cMIM (orange), cMIM-$\Sigma$ (gray) (replace expectation with sum in Eq. (4)), InfoNCE (purple), InfoNCE-X (yellow) (InfoNCE without positive augmentations). (a) cMIM-$\Sigma$ and InfoNCE-X underperform their originals. (b) Both variants are more batch-size sensitive (wider slope spread), supporting the mean-denominator design in cMIM and the importance of positives for InfoNCE. We also tested cAE/cVAE (adding the regularizer to AE/VAE) and observed no gains; see Appendix E.2.

## 3.6 Ablation

We ablate two key choices: (i) replacing the expectation in Eq. (4) with a sum (cMIM-$\Sigma$), and (ii) removing positive augmentations from InfoNCE (InfoNCE-X). Figure 6 shows both ablations reduce accuracy and increase batch-size sensitivity. Moreover, adding the contrastive regularizer to AE/VAE alone (cAE/cVAE) does not help, suggesting the benefit arises from cMIM's combination of MIM-style local attraction and global angular separation. The full ablation results appear in Appendix E.2, including cAE and cVAE.

## 4 Related Work

**Contrastive Learning.** Contrastive learning has become a cornerstone of self-supervised representation learning, with methods such as CPC van den Oord et al. (2018), SimCLR Chen et al. (2020), and MoCo He et al. (2020) demonstrating strong discriminative performance. These approaches typically rely on data augmentation to form positive pairs, making their success dependent on carefully chosen invariances. Augmentation-free contrastive methods, such as BYOL Grill et al. (2020) and SimSiam Chen & He (2021), avoid negatives but often require additional predictors or asymmetries for stability. Our work differs by integrating contrastive learning directly into a probabilistic framework, eliminating the need for augmentation or auxiliary networks.

**Mutual Information Maximization.** The Mutual Information Machine (MIM) Livne et al. (2019) and follow-up works Reidenbach et al. (2023) maximize mutual information between inputs and latent codes while encouraging latent clustering. Related approaches such as Deep InfoMax Hjelm et al. (2018) and InfoVAE Zhao et al. (2017) also maximize information-theoretic quantities, but typically lack a generative auto-encoding structure, or require various approximations and weighted losses which are hard to tune. Our method extends MIM with a contrastive component, addressing its limited discriminative power.

**Informative Embeddings.** Extracting hidden states from encoder–decoder models has proven effective in large language models Brown et al. (2020); Lee et al. (2024). Similarly, representations from intermediate layers of auto-encoders or VAEs have been used for downstream prediction tasks Alemi et al. (2018). We generalize this idea by introducing *informative embeddings*, a systematic

method to leverage decoder hidden states in probabilistic auto-encoders, demonstrating significant gains in both image and molecular tasks.

**Unifying Generative and Discriminative Learning.** Bridging generative modeling with discriminative performance has been a longstanding goal, explored in frameworks such as $\beta$-VAE Higgins et al. (2017), InfoGAN Chen et al. (2016), and hybrid likelihood–contrastive models van den Oord et al. (2018). Our work contributes to this line by showing that cMIM yields a single framework that maintains generative fidelity while significantly improving discriminative utility.

## 5 LIMITATIONS

While cMIM demonstrates clear benefits in discriminative performance and robustness to batch size, several limitations remain. First, we evaluate generative capacity primarily through reconstruction, leaving open the question of how cMIM performs on challenging generative tasks such as sample quality, diversity, likelihood estimation, or controlled generation. Second, our empirical validation is restricted to moderate-scale models and datasets; it remains to be seen how the method scales to larger architectures and high-dimensional modalities such as video or long-context language. Third, although cMIM removes the need for data augmentation, the choice of similarity function and temperature parameter $\tau$ may still influence results and require tuning. Finally, while we highlight reduced sensitivity to batch size, the method continues to benefit from larger effective numbers of negatives, which can introduce computational overhead when using memory queues or very large batches. These limitations motivate future work in scaling cMIM, expanding to more modalities, and further analyzing its generative behavior.

## 6 CONCLUSIONS

In this paper, we introduced cMIM, a contrastive extension of the MIM framework. Unlike conventional contrastive learning, cMIM does not require positive data augmentation and exhibits reduced sensitivity to batch size compared to InfoNCE. Our experiments show that cMIM learns more informative discriminative features than MIM, VAE, AE and InfoNCE, and outperforms MIM and InfoNCE in classification and regression tasks. Moreover, cMIM maintains comparable reconstruction quality to MIM, suggesting similar performance for generative applications, though further empirical validation is needed.

We also proposed a method for extracting embeddings from encoder–decoder models, termed *informative embeddings*, which improve the effectiveness of the learned representations in downstream applications.

Overall, cMIM advances the goal of unifying discriminative and generative representation learning. We hope this work provides a foundation for developing models that excel across a broad spectrum of machine learning tasks and motivates further research in this direction.

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

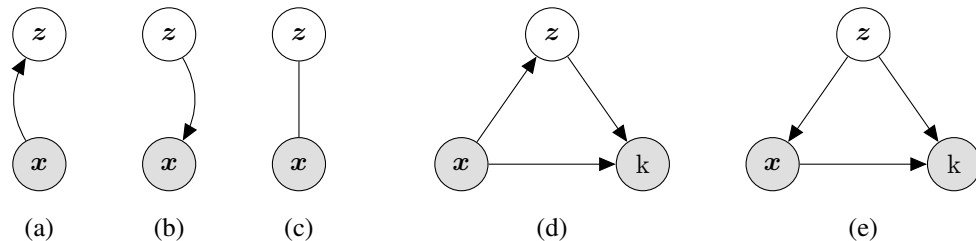

Figure 7: (Left) A MIM model learns two factorizations of a joint distribution: (a) encoding; (b) decoding factorizations; and (c) the estimated joint distribution (an undirected graphical model). (Right) We extend the MIM model with an additional binary variable k, and present the two factorizations of a joint distribution: (d) encoding; (e) decoding factorizations.

## REPRODUCIBILITY STATEMENT

We provide in Appendix D the complete details that allow reproducing our experiments, including model architectures, training hyper-parameters, and full dataset details. We also plan to release the code to reproduce all our experiments at a future date.

## ETHICS STATEMENT

**Datasets and licenses.** We use MedMNIST v2 and EMNIST/MNIST/Fashion-MNIST for images, and ZINC15 SMILES for molecules. All datasets were obtained from their official sources and used under their respective terms; we do not redistribute raw data and our code will include download scripts that point to official providers. Image datasets contain no personally identifiable information to the best of our knowledge.

**Potential misuse.** Although our molecular experiments focus on representation learning and property prediction on public benchmarks, generative models can be misused to propose harmful compounds. We do not release task-specific molecular generators; released checkpoints (if any) are intended for representation learning only. We encourage downstream users to follow domain-specific safety review, screening, and governance practices.

**Privacy and security.** The work does not involve human subjects, private data, or deployment. We adhere to the dataset maintainers' licenses and terms of use and to ICLR's Code of Ethics.

## A    LLM USAGE

We used LLM to help with polishing the writing, improving clarity, and fixing grammar issues.

## B    MIM GRAPHICAL MODEL

MIM, the Mutual Information Machine model (Livne et al., 2019) is a probabilistic auto-encoder designed to learn informative and clustered latent codes. The clustering is achieved by minimizing the marginal entropy of the latent distribution over $z$, which results in latent codes that are closely positioned in Euclidean space for similar samples (see example in the work by Reidenbach et al. (2023)). In MIM, similarity between samples is defined by the decoding distribution, leading to a local structure around each latent code (*i.e.*, similar samples correspond to nearby latent codes). However, the global distribution of these latent codes, while aligned with a target or learned prior, may not be well-suited for discriminative tasks. To address this limitation, we propose augmenting the MIM objective with a contrastive objective term, which encourages the latent codes of dissimilar samples to be more distinct from each other. This modification aims to improve the global structure of the latent space, making it more suitable for discriminative downstream tasks. See Fig. 7 for graphical model.

## C   EXTENDED FORMULATION

### C.1   ADDITIONAL NOTES ON CONTRASTIVE LEARNING

In practice, Eq. (1) implements a $B$-way classification problem where the positive is one of the $B$ candidates; performance depends on (i) the semantic validity of data augmentations defining positives, and (ii) the effective number and diversity of negatives (batch size or memory queue). These sensitivities are particularly acute for modalities where augmentations are hard to design (e.g., text).

### C.2   EXPECTATION AND BATCH-SIZE ROBUSTNESS

With cosine similarity $s\left(\cdot,\cdot\right)\in[-1,1]$ and $g\left(\cdot,\cdot\right)=\exp(s\left(/,\tau\right))\in[\mathrm{e}^{-1/\tau},\mathrm{e}^{1/\tau}]$, the in-batch Monte-Carlo estimator in Eq. (4) concentrates via Hoeffding's inequality Hoeffding (1963):

$$\Pr\left(\left|\frac{1}{B-1}\sum_{j\neq i}g\left(\boldsymbol{z}_i,\boldsymbol{z}_j\right)-\mu\right|\geq\epsilon\right)\leq 2\exp\left(-\frac{2(B-1)\epsilon^2}{(\mathrm{e}^{1/\tau}-\mathrm{e}^{-1/\tau})^2}\right). \tag{11}$$

Thus the variance is $\mathcal{O}(1/(B-1))$, explaining cMIM's robustness to batch size while still improving with more negatives.

**Conditions for the concentration bound.**   With cosine similarity $s(\cdot,\cdot)\in[-1,1]$ and fixed $\tau>0$, the random variable $g(\boldsymbol{z}_i,Z)=\exp(s(\boldsymbol{z}_i,Z)/\tau)$ is bounded in $[e^{-1/\tau},e^{1/\tau}]$. Therefore the in-batch Monte-Carlo mean $\frac{1}{B-1}\sum_{j\neq i}g(z_i,z_j)$ in Eq. (4) satisfies Hoeffding's inequality, yielding Eq (11) and variance $O(1/(B-1))$.

### C.3   DERIVATION OF THE RELATION TO INFONCE

Let $s_{ij}\triangleq s\left((,\boldsymbol{z})_i,\boldsymbol{z}_j\right)/\tau$ so that $g\left(\boldsymbol{z}_i,\boldsymbol{z}_j\right)=\exp(s_{ij})$. Starting from Eq. (4):

$$\begin{aligned}p_{k=1}&=\frac{\exp(s_{ii})}{\exp(s_{ii})+\frac{1}{B-1}\sum_{j\neq i}\exp(s_{ij})}=\frac{(B-1)\exp(s_{ii})}{(B-1)\exp(s_{ii})+\sum_{j\neq i}\exp(s_{ij})}\\&=\frac{\exp\left(s_{ii}+\log(B-1)\right)}{\exp\left(s_{ii}+\log(B-1)\right)+\sum_{j\neq i}\exp(s_{ij})}.\end{aligned} \tag{12}$$

Hence $-\log p_{k=1}$ equals an InfoNCE cross-entropy on logits $\{s_{ii}+\log(B-1),s_{ij}\ (j\neq i)\}$—InfoNCE with a fixed positive-logit offset. If the mean over negatives in Eq. (4) is replaced by the sum, the offset disappears and one recovers standard InfoNCE together with the usual $I(X;Z)\geq\log B-\mathbb{E}[\mathcal{L}_{\mathrm{InfoNCE}}]$ bound van den Oord et al. (2018). Calibration and gradient-shape remarks in the main text follow immediately from this identity.

**Calibration and gradients.**   Let $s_{ij}:=s(z_i,z_j)/\tau$ and define the log-mean-exp over negatives $\bar{s}_i\triangleq\log\left(\frac{1}{B-1}\sum_{j\neq i}e^{s_{ij}}\right)$. Eq. (4) implies

$$p_{k=1}(x_i,z_i)=\frac{1}{1+\exp(\bar{s}_i-s_{ii})}=\sigma\left(s_{ii}-\bar{s}_i\right),\qquad-\log p_{k=1}=\mathrm{softplus}\left(\bar{s}_i-s_{ii}\right).$$

Hence with $\ell_i=-\log p_{k=1}$ we obtain the closed-form gradients

$$\frac{\partial\ell_i}{\partial s_{ii}}=p_{k=1}-1,\qquad\frac{\partial\ell_i}{\partial s_{ij}}=(1-p_{k=1})\,\pi_{ij},\quad\pi_{ij}:=\frac{e^{s_{ij}}}{\sum_{l\neq i}e^{s_{il}}}.$$

*Implications.* (i) The decision boundary is the *margin* $\Delta_i=s_{ii}-\bar{s}_i=0$ against the log-mean-exp of negatives, so when all logits are equal we have $p_{k=1}=1/2$ (contrast: InfoNCE gives $1/B$). (ii) The positive gradient magnitude is $|p_{k=1}-1|$ and the negative gradient mass $(1-p_{k=1})$ is distributed *only* across negatives via $\pi_{ij}$. Together with the MIM term (local attraction), this yields angular separation calibrated by a sigmoid in the margin, while avoiding the log-sum-exp dependence on $B$ introduced by InfoNCE via Eq. (1).

## C.4 FURTHER DISCUSSION: NO-POSITIVE-AUGMENTATION REGIME

Unlike conventional contrastive methods, cMIM does not require explicit positive pairs via data augmentation: the MIM term already pulls matched $(x, z)$ pairs together (local clustering), while the $p_{k=1}$ term imposes global angular separation against the batch, simplifying training and hyper-parameter tuning. For modalities with expensive or ill-defined augmentations (e.g., text), this removes a key bottleneck.

## C.5 REMARKS ON MUTUAL-INFORMATION BOUNDS

During training, k is treated as part of the observed variable so the MI lower bound targets $I_{\mathcal{M}_S}(x, \mathrm{k}; z)$, which is equivalent to $I_{\mathcal{M}_S}(x; z)$ since $\mathrm{k} \equiv 1$. Thus cMIM inherits MIM's MI guarantees even though its contrastive calibration differs from InfoNCE and does not directly yield the classical InfoNCE MI bound.

# D EXPERIMENT TRAINING DETAILS

## D.1 IMAGE CLASSIFICATION

| # | Dataset | Train Samples | Test Samples | Categories | Description |
|---|---------|---------------|--------------|------------|-------------|
| 1 | MNIST | 60,000 | 10,000 | 10 | Handwritten digits |
| 2 | Fashion MNIST | 60,000 | 10,000 | 10 | Clothing images |
| 3 | EMNIST Letters | 88,800 | 14,800 | 27 | Handwritten letters |
| 4 | EMNIST Digits | 240,000 | 40,000 | 10 | Handwritten digits |
| 5 | PathMNIST | 89,996 | 7,180 | 9 | Colon tissue histology |
| 6 | DermaMNIST | 7,007 | 2,003 | 7 | Skin lesion images |
| 7 | OCTMNIST | 97,477 | 8,646 | 4 | Retinal OCT images |
| 8 | PneumoniaMNIST | 9,728 | 2,433 | 2 | Pneumonia chest X-rays |
| 9 | RetinaMNIST | 1,600 | 400 | 5 | Retinal fundus images |
| 10 | BreastMNIST | 7,000 | 2,000 | 2 | Breast tumor ultrasound |
| 11 | BloodMNIST | 11,959 | 3,432 | 8 | Blood cell microscopy |
| 12 | TissueMNIST | 165,466 | 47,711 | 8 | Kidney tissue cells |
| 13 | OrganAMNIST | 34,581 | 8,336 | 11 | Abdominal organ CT scans |
| 14 | OrganCMNIST | 13,000 | 3,239 | 11 | Organ CT, central slices |
| 15 | OrganSMNIST | 23,000 | 5,749 | 11 | Organ CT, sagittal slices |

Table 3: **Image Classification:** Summary of train/test samples, categories, and descriptions for MNIST, FashionMNIST, EMNIST, and MedMNIST datasets (rows 5-15).

**Dataset:** Default train and test splits were used. When default validation set was not available, 5% of train was used. See Table 3 for details.

**Data augmentation:** The usual data augmentation was used as a regularization technique during training for all models. A random affine transform was applied to all images during training with default parameters of:

- degrees=15
- translate=(0.1, 0.1)
- scale=(0.9, 1.1)
- shear=10

**Model and Architecture details:** We opted for a simple architecture.

- The encoder flattens the image to 784 dimensions, up-projects using a linear layer to $(784, 16)$ which is fed to a Perceiver encoder that projects it down to 400 steps $(400, 16)$. A linear layer projects the hidden dimension to 1, followed by a layer norm, and finally a linear projection from 400 to 64.

- The encoding distribution is a Gaussian with mean and variance predicted by linear layers from the encoder output.
- The decoder up-projects the 64 dimension latent code using a linear layer to $(64, 16)$ which is fed to a Perceiver encoder that projects it down to 400 steps $(400, 16)$. A linear layer projects the hidden dimension to 1, followed by a layer norm, and finally a linear projection from 400 to 784, which is reshaped back to $(28, 28)$ image dimensions.
- The decoding distribution is a conditional Bernoulli with logits predicted by a linear layer from the decoder output.
- The prior is a standard Gaussian.

**Optimization:** All models were trained with Adam optimizer with learning rate $1e - 3$ and WSD scheduler with 10% warmup steps and 10% decay steps, for a total of 1M steps (regardless of the batch size).

**Classification:** We report results using KNN (cosine and Euclidean) and a one-hidden-layer MLP with 400 dimensions. We use Scikit-learn Pedregosa et al. (2011) with default values.

### D.2 MOLECULAR PROPERTY PREDICTION

**Dataset:** All models were trained using a tranche of the ZINC-15 dataset (Sterling & Irwin, 2015), labeled as reactive and annotated, with molecular weight $\leq 500$Da and logP $\leq 5$. Of these molecules, 730M were selected at random and split into training, testing, and validation sets, with 723M molecules in the training set, out of which 100k molecules were used as the validation set, and 7M molecules in the testing set. We note that we do not explore the effect of model size, hyper-parameters, and data on the models. Instead, we train all models on the same data using the same hyperparameters, focusing on the effect of the learning framework and the fixed-size bottleneck. For comparison, Chemformer was trained on 100M molecules from ZINC-15 (Sterling & Irwin, 2015) – 20X the size of the dataset used to train CDDD (72M from ZINC-15 and PubChem (Kim et al., 2018)). MoLFormer-XL was trained on 1.1 billion molecules from the PubChem and ZINC datasets.

**Data augmentation:** Following Irwin et al. (2022), we used two augmentation methods: masking, and SMILES enumeration (Weininger, 1988). Masking is as described for the BART MLM denoising objective, with 10% of the tokens being masked, and was only used during the training of MegaMolBART. In addition, MegaMolBART, Perceiver AE, and MolVAE used SMILES enumeration where the encoder and decoder received different valid permutations of the input SMILES string. MolMIM was the only model to see an increase in performance when both the encoder and decoder received the same input SMILES permutation, simplifying the training procedure.

**Model and Architecture details:** We implemented all models with NeMo Megatron toolkit (Kuchaiev et al., 2019). We used a RegEx tokenizer with 523 tokens (Bird et al., 2009). All models had 6 layers in the encoder and 6 layers in the decoder, with a hidden size of 512, 8 attention heads, and a feed-forward dimension of 2048. The Perceiver-based models also required defining K, the hidden length, which relates to the hidden dimension by $H = K \times D$ where $H$ is the total hidden dimension, and $D$ is the model dimension (Fig. 1). MegaMolBART had $58.9M$ parameters, Perceiver AE had $64.6M$, and MolVAE and MolMIM had $65.2M$. We used greedy decoding in all experiments. We note that we trained MolVAE using the loss of $\beta$-VAE (Higgins et al., 2017) where we scaled the KL divergence term with $\beta = \frac{1}{D}$ where $D$ is the hidden dimensions.

**Optimization:** We use ADAM optimizer (Kingma & Ba, 2015) with a learning rate of 1.0e-4, betas of 0.9 and 0.999, weight decay of 0.0, and an epsilon value of 1.0e-8. We used Noam learning rate scheduler (Vaswani et al., 2017) with a warm-up ratio of 0.008, and a minimum learning rate of 1e-5. During training, we used a maximum sequence length of 512, dropout of 0.1, local batch size of 256, and global batch size of 16384. All models were trained for 250k steps with fp16 precision for 40 hours on 4 nodes with 16 GPU/node (Tesla V100 32GB). MolVAE was trained using $\beta$-VAE (Higgins et al., 2017) with $\beta = \frac{1}{D}$ where $D$ is the total number of hidden dimensions. We have found this choice to provide a reasonable balance between the rate and distortion (see Alemi et al. (2018) for details). It is important to note that MolMIM does not require the same $\beta$ hyperparameter tuning as done for VAE, making it easier to use in practice. The compute budget that was used is identical to the experimental seup by Reidenbach et al. (2023).

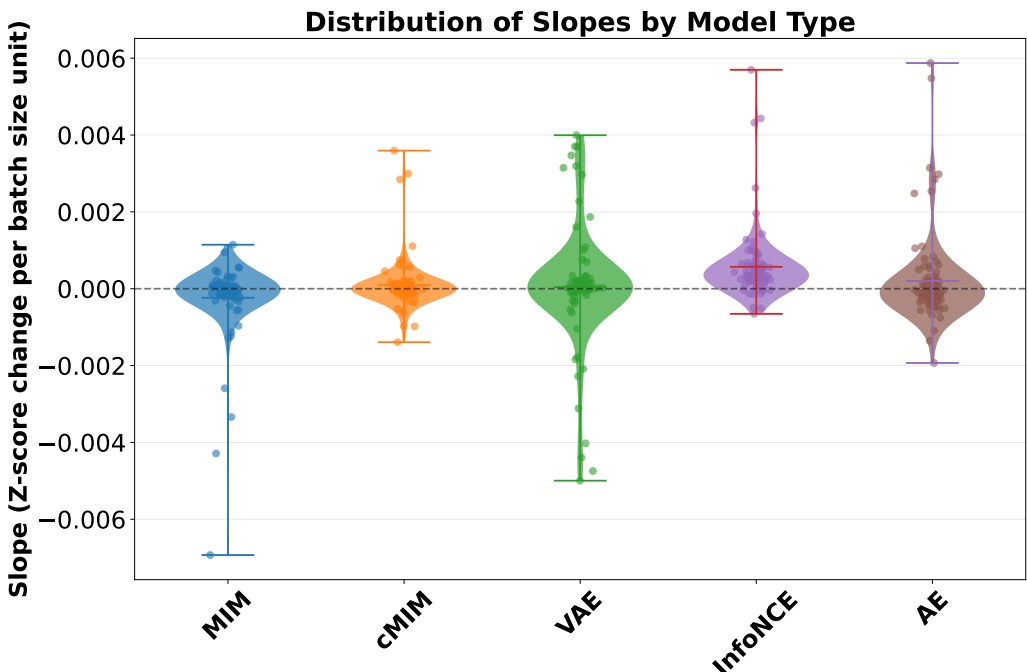

Figure 8: Distribution of slopes from linear fits of accuracy vs. batch size for different models, datasets, and evaluation metric. Each point corresponds to z-score of a model trained on MNIST-like datasets. Statistics was computed over 90 experiments (6 eval settings × 15 datasets).

**Regression:** We trained SVM and MLP regressors using BioNemo (St John et al., 2024) with default hyperparameters. MLP classifiers had one hidden layer with 128 units, ReLU activation, batch size of 32, learning rate of 1e-3, and were trained for 10000 steps. Loss was mean squared error to target values. We use SVM regressors from Scikit-learn Pedregosa et al. (2011) with default values.

## E    ADDITIONAL RESULTS

### E.1    DETAILED BATCH SIZE SENSITIVITY

In Fig. 4 in the main body we showed the slope distribution over average z-score. This provided a clean and easy to digest plot. Here we provide the distribution of each of the 90 experiments we performed on the MNIST-like data. In Fig. 8 we show the detailed slope distribution for the main models. In Fig. 9 we show the detailed slope distribution for all models we tested. cMIM is the model with the smaller spread, while being centered roughly around 0, visualizing the robustness to batch size.

### E.2    MNIST-LIKE IMAGE CLASSIFICATION ADDITIONAL RESULTS

In this section we provide additional results for MNIST-like image classification tasks. Fig. 14 shows z-scores with error bars for different evaluation methods, while Fig. 15 presents rankings with error bars for the same evaluation methods. These figures complement the main results presented in Fig. 3 of the main text by showing all models ew have tested in a single figure. We present here results for regular embeddings (Figs. 10-11), informative embeddings (Figs. 12-13), and over both methods (Figs. 14-15).

864
865
866
867
868
869
870
871
872
873
874
875
876
877
878
879
880
881
882
883
884
885
886
887
888
889
890
891
892
893
894
895
896
897
898
899
900
901
902
903
904
905
906
907
908
909
910
911
912
913
914
915
916
917

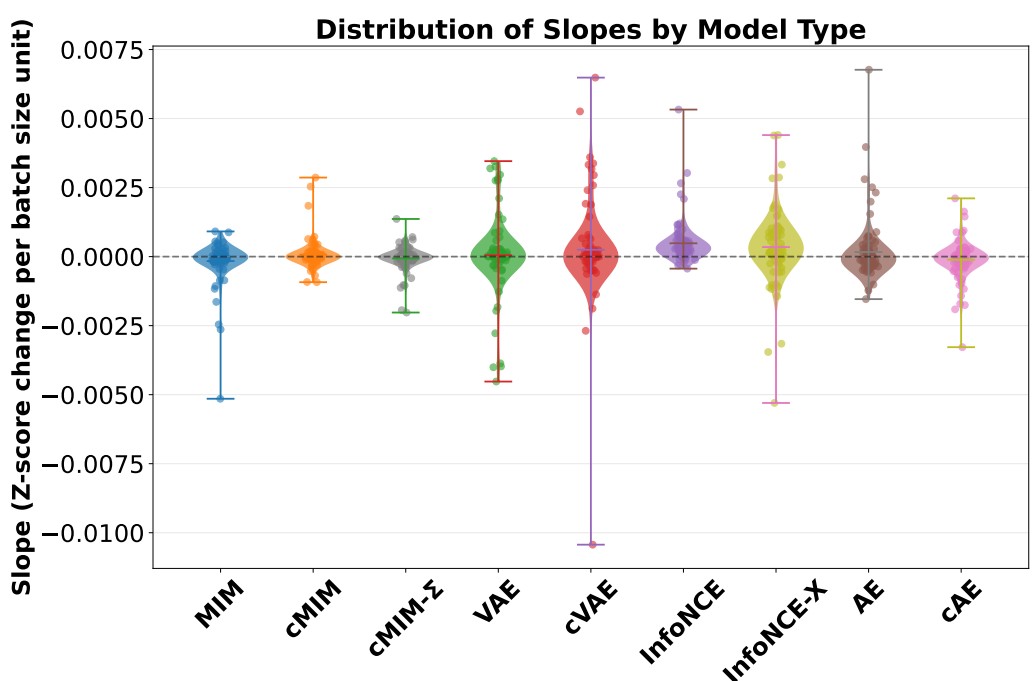

Figure 9: For completeness we provide a joint plot of all models we tested. Here we show the distribution of slopes from linear fits of accuracy vs. batch size for different models, datasets, and evaluation metric. Each point corresponds to z-score of a model trained on MNIST-like datasets. Statistics was computed over 90 experiments (6 eval settings × 15 datasets).

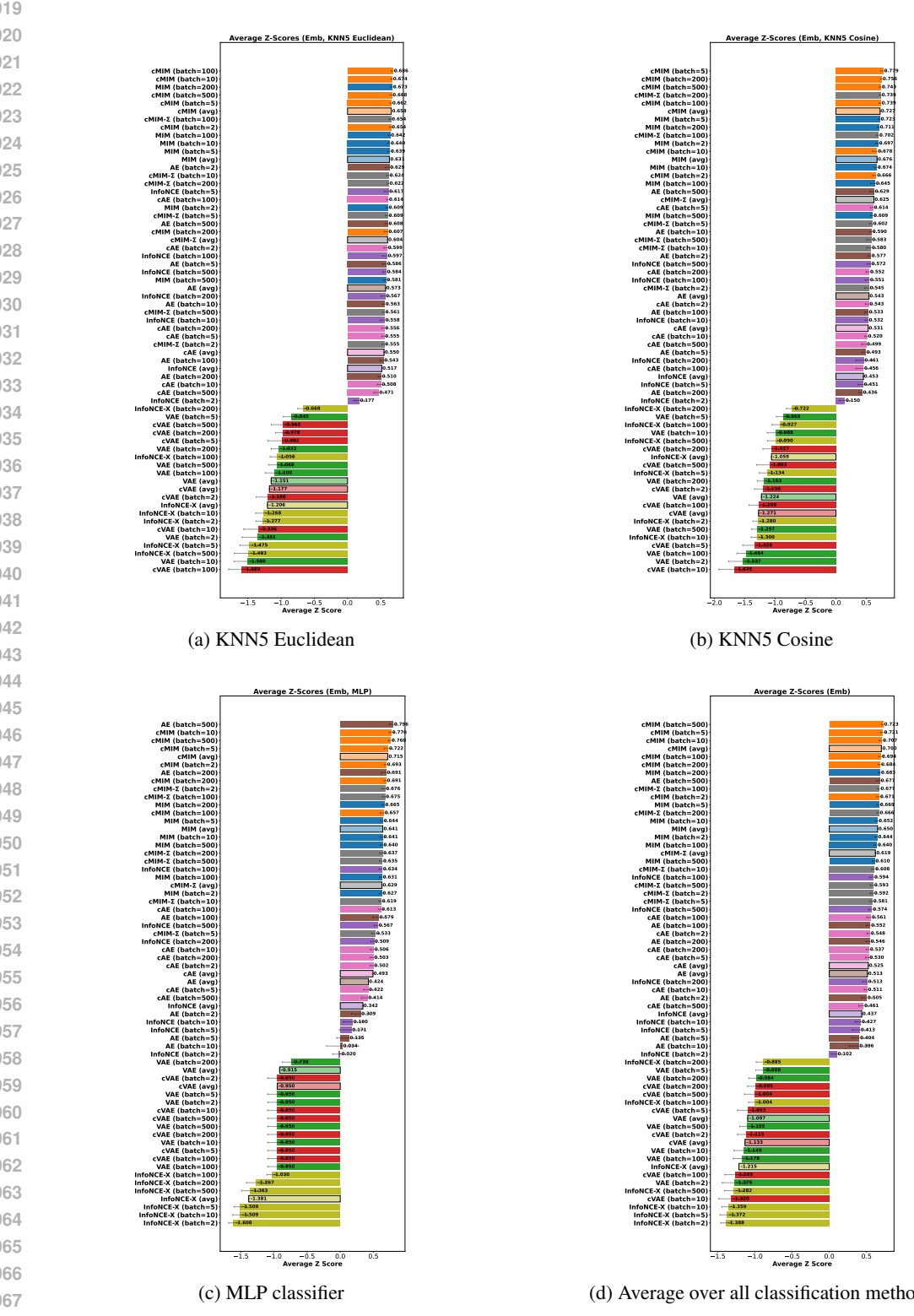

(a) KNN5 Euclidean

(b) KNN5 Cosine

(c) MLP classifier

(d) Average over all classification methods

Figure 10: Z-scores with error bars for MNIST-like image classification tasks using different evaluation methods over regular embeddings.

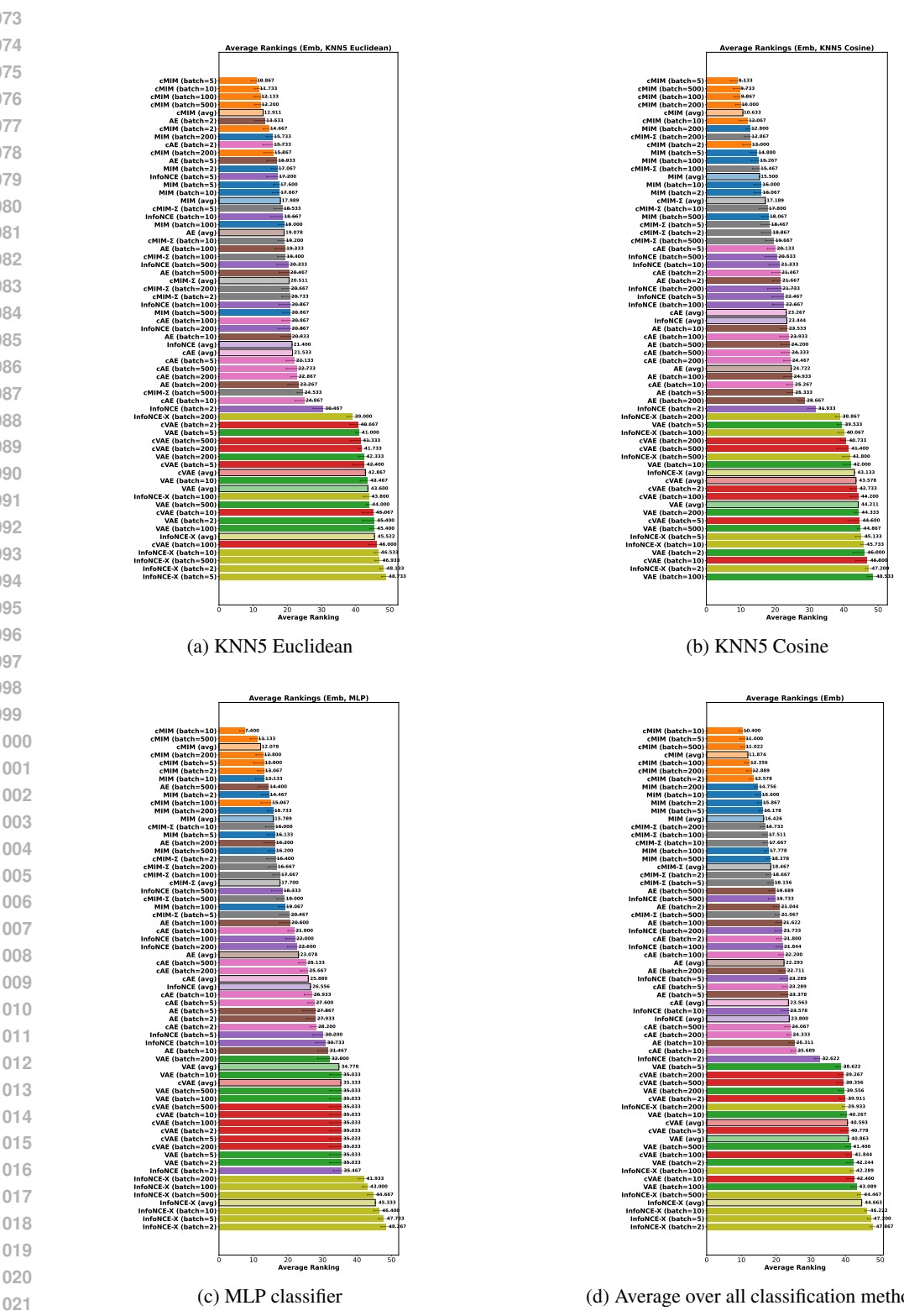

(a) KNN5 Euclidean

(b) KNN5 Cosine

(c) MLP classifier

(d) Average over all classification methods

Figure 11: Rankings with error bars for MNIST-like image classification tasks using different evaluation methods over regular embeddings.

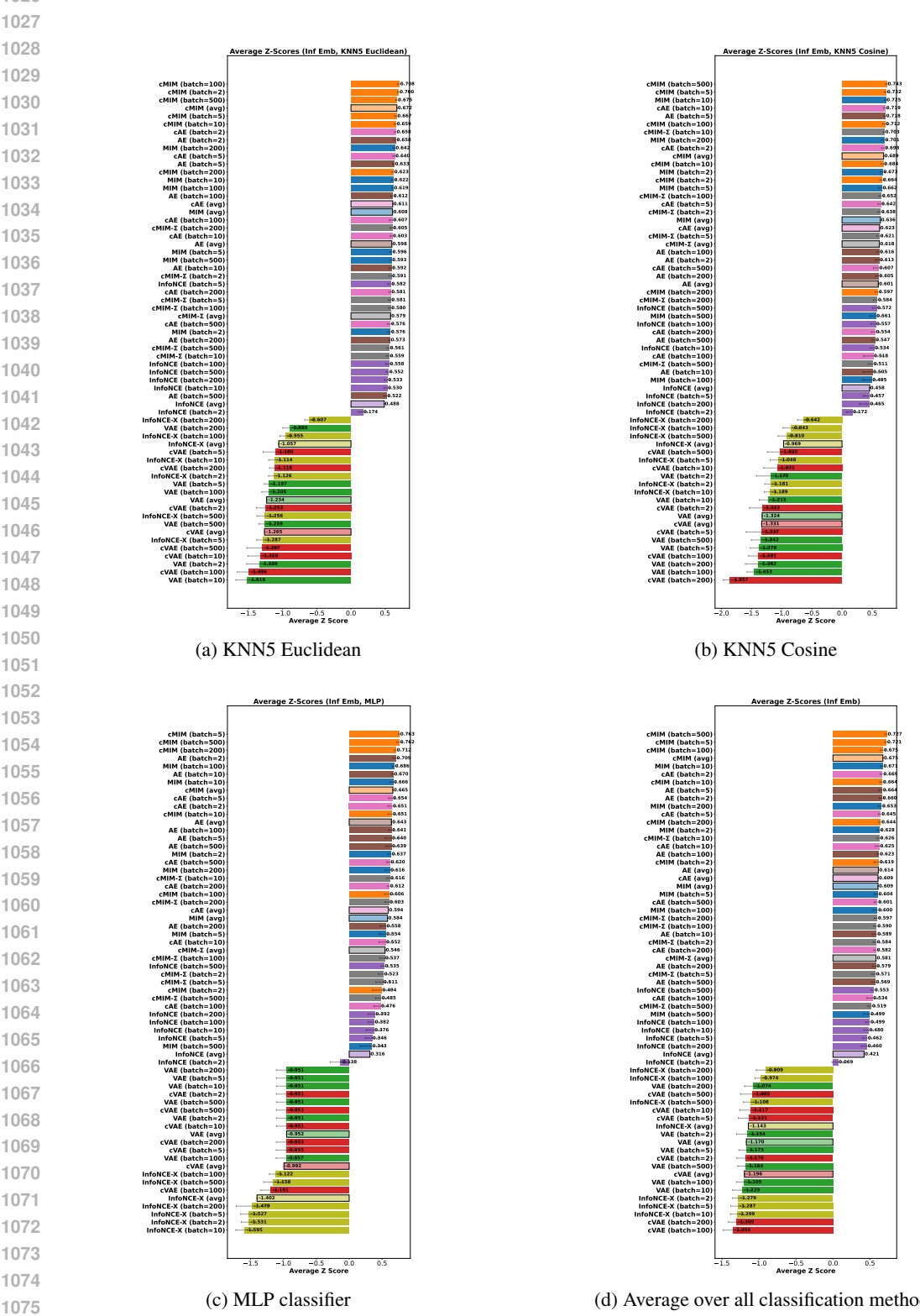

Figure 12: Z-scores with error bars for MNIST-like image classification tasks using different evaluation methods over informative embeddings.

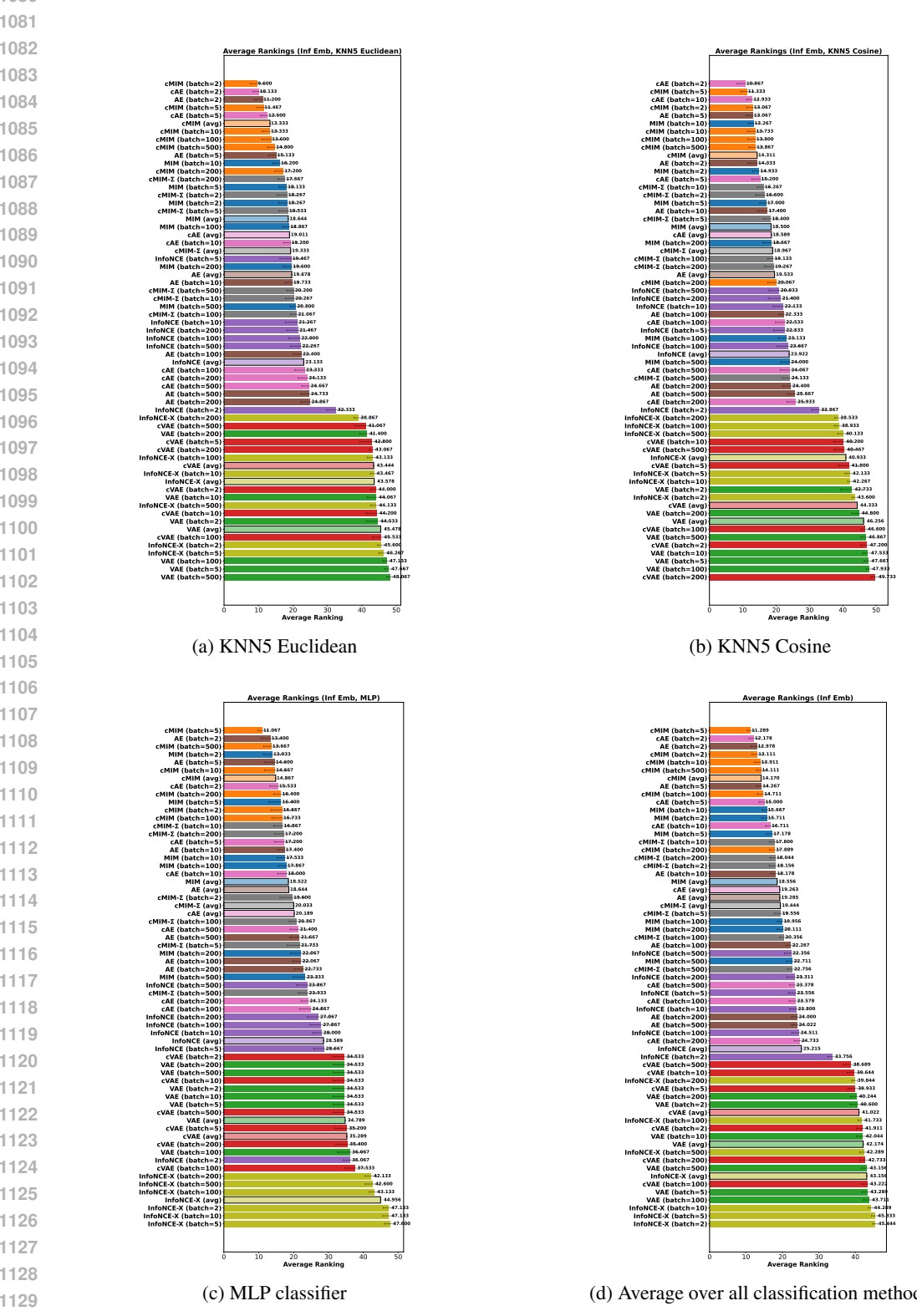

(a) KNN5 Euclidean

(b) KNN5 Cosine

(c) MLP classifier

(d) Average over all classification methods

Figure 13: Rankings with error bars for MNIST-like image classification tasks using different evaluation methods over informative embeddings.

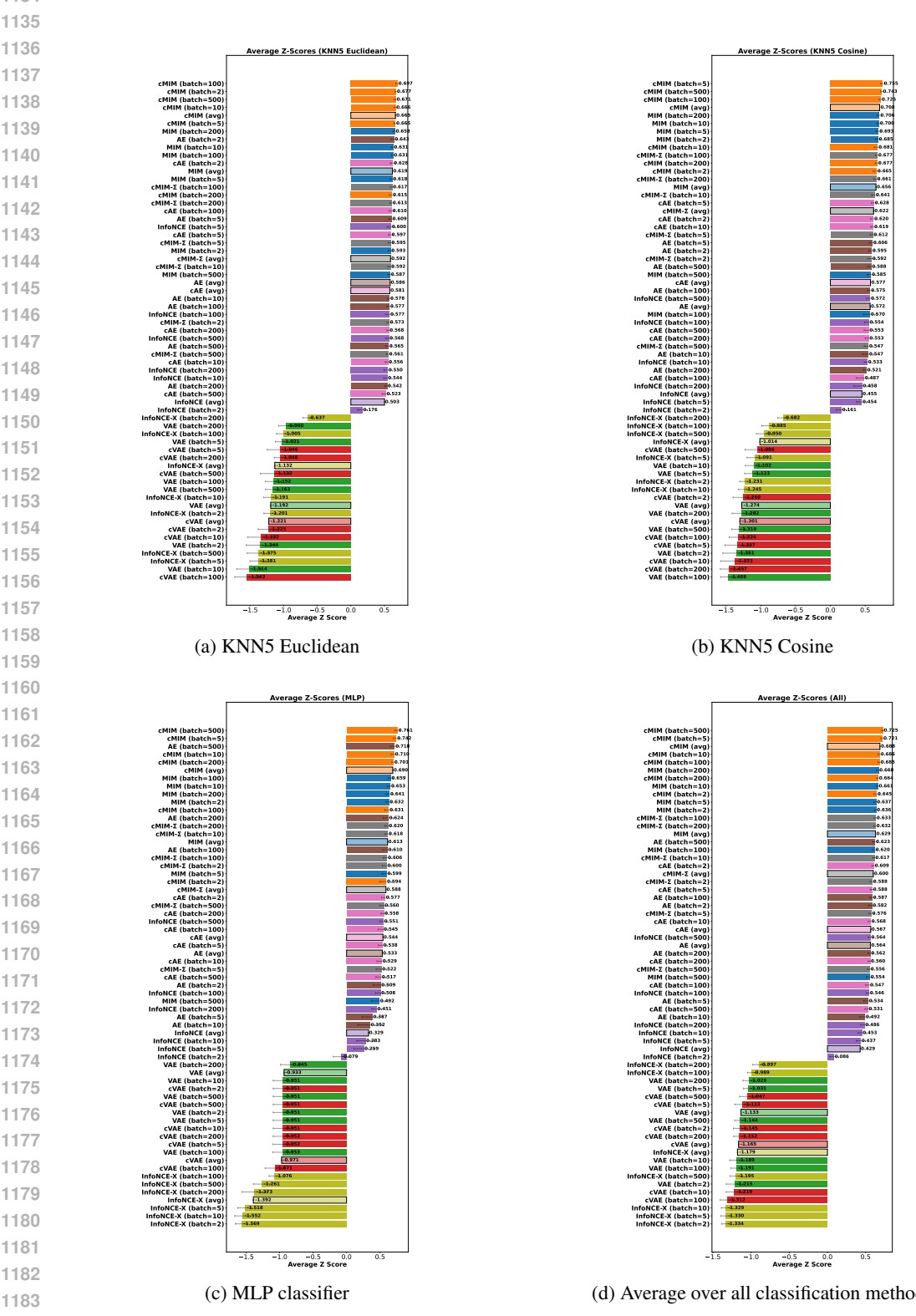

(a) KNN5 Euclidean

(b) KNN5 Cosine

(c) MLP classifier

(d) Average over all classification methods

Figure 14: Z-scores with error bars for MNIST-like image classification tasks using different evaluation methods over regular and informative embeddings.

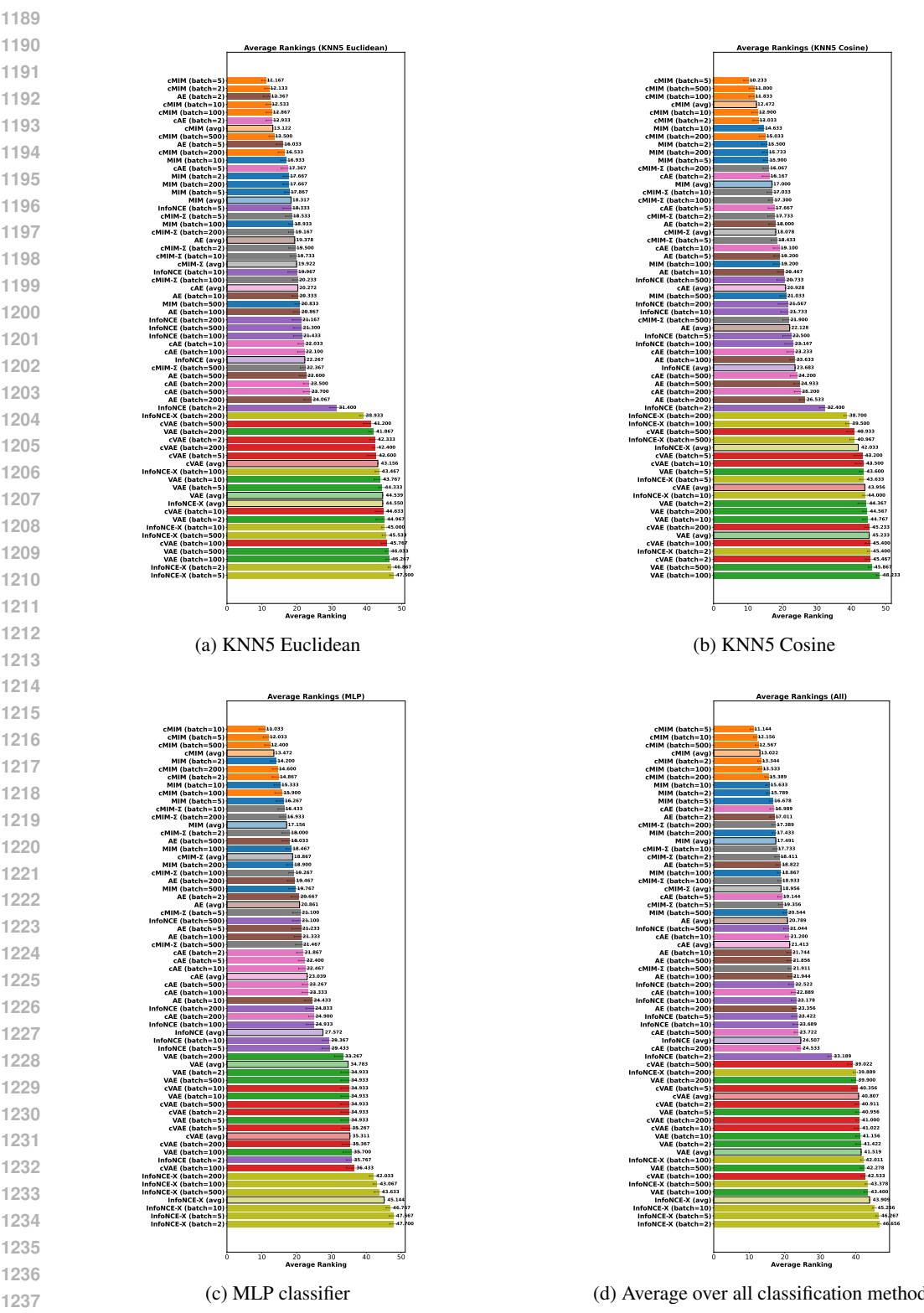

(a) KNN5 Euclidean

(b) KNN5 Cosine

(c) MLP classifier

(d) Average over all classification methods

Figure 15: Rankings with error bars for MNIST-like image classification tasks using different evaluation methods over regular and informative embeddings.

