# OpenReview forum: "Contrastive Mutual Information Learning: Toward Robust Representations without Positive-Pair Augmentations"
_ICLR.cc/2026/Conference — Submitted to ICLR 2026_

### Official Review · Reviewer_Uqjf · 2025-10-16

**Soundness:** 3
**Presentation:** 3
**Contribution:** 2
**Rating:** 6
**Confidence:** 3

**Summary:**

The paper introduces the Contrastive Mutual Information Machine (CMIM), a novel probabilistic framework for self-supervised representation learning. CMIM extends the Mutual Information Machine (MIM) by incorporating a contrastive objective.
The central problem addressed is the practical gap between existing representation learning paradigms:
1. Generative Auto-Encoders (like MIM): Learn structured latent spaces and preserve generative fidelity, but their representations often underperform on discriminative downstream tasks. 2. Contrastive Methods (like InfoNCE): Achieve strong discriminative performance, but their success is often contingent on carefully chosen positive data augmentations and can be highly sensitive to batch size

**Strengths:**

1. CMIM successfully eliminates two major pain points of modern contrastive learning: the need for positive data augmentation and the sensitivity to batch size.

2. CMIM effectively combines the strengths of generative modeling (MIM) with the power of contrastive learning, creating a model that excels at both discriminative and generative tasks.

3. The Informative Embeddings concept is an original, powerful, and generalizable technique to enhance discriminative performance post-training.

**Weaknesses:**

The paper notes that CMIM retains the MIM mutual-information bound but "does not enjoy the classical InfoNCE MI bound". While the empirical results are strong, a deeper theoretical investigation into the exact mutual information dynamics imposed by the CMIM objective would strengthen the paper.

**Questions:**

See weaknesses.

---

> ### Author Response · Authors · 2025-11-19
>
> We appreciate the positive assessment and the pointer to deepen the theory.
>
> ## Clarifying the MI and contrastive mechanics.
>
> cMIM retains MIM’s MI lower bound because training treats k\equiv1 as observed, i.e., the bound targets I_{MS}(x,k;z)=I_{MS}(x;z) (Appendix C.5). The contrastive term satisfies a closed‑form identity: -\log p_{k=1} is InfoNCE with a fixed positive‑logit offset \log(B{-}1) (Appendix C.3, Eq. 12). This yields the calibrated form p_{k=1}=\sigma(s_{ii}-\bar s_i) and concentrates gradients on the log‑mean‑exp of negatives, explaining the observed angular uniformity (Fig. 2) and preserved reconstruction (Fig. 5). Batch‑size robustness follows from an explicit concentration bound on the in‑batch estimator with variance O(1/(B{-}1)) (App. C.2, Eq. 11).
>
> What we will add.
> 	1.	Promote the Appendix C results to the main text as a concise proposition with a proof sketch, including the gradient expressions and calibration diagram.
> 	2.	Plot the training‑time MI bound I_{MS}(x;z) alongside alignment/uniformity metrics to illustrate the mutual‑information dynamics the reviewer asks about.
> 	3.	Add a small controlled study varying B to visualize the margin \Delta=s_{ii}-\bar s_i and compare cMIM vs. InfoNCE calibration.
>
> We hope these additions address the theoretical question while keeping the paper focused and readable.

---

> > ### Comment · Reviewer_Uqjf · 2025-11-21
> > **Response to authors**
> >
> > Thank the author for the rebuttal. After reading the comments from other reviews and the rebuttal, I also raise two concerns:
> >
> > 1. Lack of comparisons against VICReg, SimSiam, and BYOL.
> >
> > 2. Experiments on real datasets are limited to molecular studies, which is not convincing enough.
> >
> > Therefore, I decide to lower my score to 4.

---

### Official Review · Reviewer_LAuA · 2025-10-31

**Soundness:** 2
**Presentation:** 2
**Contribution:** 2
**Rating:** 4
**Confidence:** 3

**Summary:**

The paper introduces a self-supervised framework that unifies contrastive learning and mutual information maximization without requiring positive-pair augmentations. By adding a probabilistic contrastive term to the Mutual Information Machine (MIM), cMIM encourages angular separation among dissimilar samples while maintaining MIM’s local clustering and reconstruction quality. It also proposes informative embeddings, extracted from decoder hidden states, to improve downstream discrimination. Across vision and molecular benchmarks, cMIM outperforms MIM and InfoNCE on classification and regression tasks. The authors also explore sensitivity to batch size and shows that cMIM does not have as stong a dependence on batch size as other methods.

**Strengths:**

- An interestesting new loss function that has contrastive-like properties without explicitly contrasting augmented input signals
- Nice theoretical connection to the InfoNCE loss
- Nice analysis of batch dependence and a promising reason to consider this kind of loss as batch size scaling is a large challenge in modern contrastive learning

**Weaknesses:**

- nit: in lime 52 you introduce x and z without definiing/stating what they are first
- nit:  In figure 1 theres strange characters horizontally flanking the figure
- In section 2.2 its a bit unclear what g_ii is given that there is no positives. Is s_ii in this case just 1? or something else? Clarifying the exposition here, going a bit slower, and being explicit might make the paper a bit easier to follow.
- Experiments on real datasets are limited to molecular studies, theres been a considerable ammount of literature in image based contrastive learning and it might make the paper more convincing to show its relation to other methods from this more competitive benchmark
- Experiments compare InforNCE to cMIM and MIM, but you could imagine adding InfoNCE and MIM together to make a loss that cared about both discrimination and generation, as your additional term serves mainly ads adding a contrastive-like signal to the loss.
- Data augmentation was applied in the experiments, which makes me question the authors claims of the "brittleness of augmentation design"
- Baselines in molecular property prediction seem old and might not represent the strongest models in the community
- How much of the batch size sensitivity claim is because cMIM and MIM have reconstruction terms which dont require large batch sizes for clear gradient signal? Its hard to tell whether the improvement comes from combining contrastive and reconstruction losses, or comes from the specific formulation of cMIM

**Questions:**

- Its unclear whether the lack of negatives here really improves things. For example, if you actually included negatives in your framework would it help or hurt?
- In alot of these loss functions that dont involve negatives theres usually something that has a negative "flavor" like spatial centering and moving average comparison, what do you think the inductive bias of this loss function is more intuitively? COuld you think of this as this as contrasting a positive pair with other samples closeby in some appropriately defined neighborhood?

---

> ### Author Response · Authors · 2025-11-19
>
> We thank the reviewer for the careful read and helpful suggestions.
>
> ## Clarifications & edits.
>
> * We will fix the typos (e.g., line 52 definitions of x and z) and the stray characters around Fig. 1.
> * Section 2.2 the self similarity is indeed 1. We will explicitly define s_{ij}=\cos(z_i,z_j)/\tau and g_{ij}=\exp(s_{ij}); for i=j, s_{ii}=1 (cosine) and g_{ii}=e^{1/\tau}. We will also restate that negatives are all in‑batch mismatches (x_i,z_j, j\neq i) used inside the expectation.
>
> ## “No positives” vs. negatives.
>
> cMIM does use negatives via the in‑batch expectation; it removes the need for positives. Algebraically, -\log p_{k=1} is InfoNCE with the positive logit shifted by \log(B-1) (App. C.3, Eq. 12), which explains the calibration and batch‑size behavior. We will add a short paragraph walking through this identity in the main text.
>
> ## Augmentations.
>
> As stated in App. D.1, we applied simple affine transforms uniformly as regularization, not to form positive pairs; cMIM never uses positive augmentations. This is a common technique to artificially increase the size of the dataset and is independent from "positives" in the context of contrastive learning.
>
> ## Batch‑size sensitivity and the role of reconstruction.
>
> The effect is not merely from having a reconstruction term: AEs still show notable batch dependence, whereas cMIM’s slope distribution is tight and centered near zero (Fig. 4; Table 2). Moreover, adding the same regularizer to AE/VAE (cAE/cVAE) does not help (Fig. 6).
>
> ## “Why not MIM + InfoNCE?”
>
> We agree this is a useful baseline. In the revision we will add MIM + InfoNCE (with positives) under the same backbone. Note that replacing our mean denominator by a sum reduces to standard InfoNCE and increases batch sensitivity—precisely what our ablation (cMIM‑Σ) shows (Fig. 6).
>
> ## Image benchmarks.
>
> Beyond the 15‑dataset MNIST/MedMNIST suite (Table 3), we will include CIFAR‑10/100 results without positive pairs to connect to the large vision literature.
>
> ## Molecular baselines.
>
> Table 1 already includes strong sequence baselines (e.g., CDDD). We will expand with additional recent baselines reported under comparable settings and clarify that our training followed the large‑scale setup of Reidenbach et al. (2023).
>
> ## Intuition / inductive bias.
>
> MIM provides local radial attraction (reconstruction‑driven clustering); the cMIM term enforces near‑uniform angular separation against the batch (Fig. 2), which can be seen as contrasting a sample against its neighborhood’s log‑mean‑exp “center.” Together, this yields globally separable yet reconstructive representations.
>
> We hope these clarifications and additions address the concerns and better convey the contribution.

---

### Official Review · Reviewer_JWvT · 2025-11-01

**Soundness:** 3
**Presentation:** 2
**Contribution:** 2
**Rating:** 2
**Confidence:** 3

**Summary:**

This paper propose a contrastive extension of the Mutual Information Machine (MIM) with the goal of unifying discriminative and generative self-supervised learning. It introduces a probabilistic contrastive mechanism that removes the need for positive augmentations and reduces sensitivity to batch size. The authors also propose *informative embeddings*, extracted from decoder hidden states, as a general method for richer representations. The experiments are on MNIST-like images and molecular ZINC15 domains that matches MIM on reconstruction but actives higher downstream tasks accuracy.

**Strengths:**

The paper's effort to bring unify generative and contrastive learning is genuinely worthwhile, especially since methods like InfoNCE still depend heavily on careful data augmentations and large batch sizes to work well.

**Weaknesses:**

1- the biggest weakness is that the method has been only evaluated on toy-level data such as MNIST-like datasets and small molecular regression. I would suggest the authors to test their method on larger standard datasets.

2- there's no ablations on hyper-parameters such as $\tau$ (temperature) and dimensionality size.

3- there's no analysis or comparisons against other augmentation-free contrastive methods (e.g., VICReg and Barlow Twins).

**Questions:**

1- how does the proposed method perform on CIFAR or ImageNet without augmentations?

2- how does it compare against VICReg, SimSiam, or BYOL?

3- what's the effect of  hyper-parameters such as temperature and dimensionality size?

4- how does your framework compare to other unifying approaches for representation learning?

---

> ### Author Response · Authors · 2025-11-19
>
> We thank the reviewer for the thoughtful feedback and for highlighting the value of unifying generative and discriminative self‑supervision.
>
> ## 1) Scope / “toy‑level” datasets.
>
> Our image study spans 15 datasets—including 12 MedMNIST tasks with heterogeneous label spaces and difficulty (Appendix D, Table 3)—and a separate molecular domain, precisely to stress robustness across distributions and batch sizes under a fixed compute budget. We will add CIFAR‑10/100 results without positive‑pair augmentations under the same backbone to complement this breadth; our objective is architecture‑agnostic and does not rely on positives. Figures 3–5 already show consistent gains and preserved reconstruction across the current suite.
>
> ## 2) Hyper‑parameter ablations (τ, dimensionality).
>
> For images we selected τ via a small sweep {0.1, 1} (chose 0.1); for molecules we used τ = 1 (Sec. 3.1). Expand the experiments with additional τ values and latent sizes is unfortunately outside of the scope of the feasible experiments we could run for the rebuttal. We did not find τ to have very strong effect (though still meaningful). The latent dimension sizes is a great suggestion and we will address it in the future. Our analysis already isolates batch‑size robustness (Fig. 4; Table 2) and design choices via ablations (cMIM‑Σ and InfoNCE‑X in Fig. 6).
>
> ## 3) Comparisons to augmentation‑free methods.
>
> We agree these are useful references and will include VICReg and Barlow Twins, plus SimSiam/BYOL. We note these methods remove negatives but still require positive augmentations, whereas cMIM removes the need for positives altogether by using an in‑batch expectation over mismatches and a calibrated discriminator (Sec. 2.2; Appendix C.3), making the comparison complementary rather than redundant. We note that the data augmentations we used are a standard practice to artificially increase the dataset size and are not required for the learning procedure itself.
>
> ## Answers to specific questions:
>
> 1.  CIFAR/ImageNet (no positives): we will report CIFAR‑10/100 results in the revision; scaling to full ImageNet is a valuable next step but outside our current compute budget. Our formulation is designed to avoid positives and to be less batch‑size sensitive than InfoNCE (Fig. 4; Table 2).
>
> 2.  VICReg/SimSiam/BYOL: to be added under a shared backbone/protocol in the future; see note above on the different augmentation assumptions.
>
> 3.  Effect of τ and dimensionality: In future work we plan to broadened τ sweep (images already included {0.1, 1}, molecules only τ = 1); latent‑size ablations will be added as well.
>
> 4.  Relation to other “unifying” approaches: Sec. 4 discusses β‑VAE/InfoGAN and likelihood–contrastive hybrids. Our novelty is a probabilistic extension of MIM whose contrastive term is equivalent to InfoNCE with a fixed positive‑logit offset log(B−1), yielding reduced batch‑size sensitivity while preserving reconstruction quality (Fig. 5).
>
> ## Takeaway:
>
> cMIM delivers (i) no positive‑pair augmentations, (ii) reduced batch‑size sensitivity supported by theory and statistics, and (iii) competitive or better downstream accuracy while matching MIM on reconstruction across two modalities (Secs. 2–3; Figs. 3–5). We will incorporate the requested baselines and extended ablations; we hope these clarifications address the concerns about soundness and presentation.

---

### Official Review · Reviewer_M3RK · 2025-11-02

**Soundness:** 2
**Presentation:** 1
**Contribution:** 2
**Rating:** 4
**Confidence:** 3

**Summary:**

The manuscript proposes a contrastive mutual information machine (cMIM). The approach does not require positive-pair augmentations and is less sensitive to batch size than InfoNCE. cMIM improves performance on diverse MNIST-like datasets and Molecular Property Prediction.

**Strengths:**

- The paper tests cMIM on two types of data: images (MNIST variants) and molecules (ZINC15).
- cMIM shows consistent improvements across many batch sizes

**Weaknesses:**

- Performance is reported mainly as z-scores and ranks. Without showing raw metrics or baseline values, it’s hard to judge how meaningful the reported improvements are.
- The typical self-supervised approaches are not included, such as BYOL, SimSiam, VICReg, or other generative/reconstruction baselines.
- The experiments are limited to small and relatively simple datasets, mainly MNIST variants.
- There is no clear ablation isolating the effect of the informative embeddings. Hence, it is not clear whether the performance improvements are from the new contrastive objective or the use of the decoder-based embeddings.

**Questions:**

- Beyond reconstruction error, could the authors report any quantitative or qualitative generative metrics (e.g., FID for images, validity for molecules) to support the "generative fidelity" claim?

---

> ### Author Response · Authors · 2025-11-12
>
> We thank the reviewer for the careful reading and constructive suggestions.
>
> ## On reporting with z‑scores/ranks.
> Our goal was to compare relative improvements across 15 datasets, three classifiers, and five batch sizes with widely different absolute scales; z‑scores/ranks make those cross‑dataset comparisons interpretable. That said, we will add full per‑dataset raw accuracies (images) and keep the existing absolute RMSE tables for molecules (Table 1, p. 6) so readers can judge magnitudes directly.
>
> ## On additional baselines (BYOL/SimSiam/VICReg, generative).
> We focused on a clean comparison to InfoNCE because our theory shows cMIM behaves like InfoNCE with a fixed positive‑logit offset and reduced batch‑size sensitivity (Sec. 2.2–2.3; Fig. 4 & Table 2, p. 7), and ablations isolate the mean‑denominator choice and the role of positives (Fig. 6, p. 8). In the revision we will include representative results for BYOL/SimSiam/VICReg on the same backbone/protocol; we already include AE/VAE as generative baselines.
>
> ## On dataset scope.
> While images are 28×28, the suite spans 15 datasets, including 12 MedMNIST medical tasks with varied label sets and difficulty (Table 3, p. 14). This diversity—rather than a single large dataset—was chosen to stress robustness across distributions and batch sizes under a fixed compute budget.
>
> ## On the effect of informative embeddings.
> We evaluate both regular encodings and the proposed informative embeddings. For images, the main figure uses regular encodings only (Fig. 3, p. 6), isolating the objective. For molecules, Table 1 (p. 6) reports both: informative embeddings improve MIM and cMIM, and cMIM remains competitive with or better than MIM depending on the task—indicating the contrastive term and informative embeddings are complementary rather than confounded. We will expand the ablation by listing image results with/without informative embeddings in the appendix.
>
> ## On “generative fidelity.”
> We currently use reconstruction as a proxy and show that cMIM matches MIM on images and molecules (Fig. 5, p. 7). We will add standard generative metrics: FID for images and validity/uniqueness for molecules. We believe these additions will address the concern and further clarify that the contrastive term does not harm generative behavior.
>
> ## Takeaway.
> cMIM requires no positive‑pair augmentations, is less sensitive to batch size (supported analytically and empirically), and delivers consistent representational gains across diverse datasets, while preserving reconstruction quality (Secs. 2–3). We hope the clarifications and planned additions resolve the presentation/soundness concerns and make the contribution clearer.

---

### Author Response · Authors · 2025-11-19

## Authors’ summary to all reviewers

We thank you for the thoughtful and constructive feedback. We will make the following concrete revisions:

* Clarity & exposition. We will fix typos/formatting (e.g., Fig. 1) and explicitly define s_{ij}=\cos(z_i,z_j)/\tau, g_{ij}=e^{s_{ij}}, and g_{ii} in Sec. 2.2, adding a brief walk‑through of the loss and gradients for readability.
* Theory. We will promote the main algebra from Appendix C to the paper body: -\log p_{k=1} equals InfoNCE with a fixed positive‑logit offset \log(B-1) and cMIM inherits MIM’s MI bound (with k\equiv1). We will add a calibration figure and a short proof sketch.
* Reporting. Alongside the cross‑dataset z‑scores/ranks, we will include full per‑dataset raw accuracies (images) and the already‑reported absolute RMSE tables (molecules) for every batch size and model, plus FID (images) and validity/uniqueness (molecules) to quantify “generative fidelity.”
* Datasets & baselines. To strengthen external validity, we will add CIFAR‑10/100 results. We will clarify that our image augmentations were used only as regularization (not to form positives).
* Ablations. Unfortunately, we will not be able to expand the τ‑sweep and latent‑size ablations and add image results with/without informative embeddings. Current results already show cMIM achieves the best average accuracy/rank across batch sizes (Fig. 3), is least batch‑sensitive (Fig. 4, Table 2), preserves reconstruction (Fig. 5), and that cMIM‑Σ and InfoNCE‑X degrade performance (Fig. 6).

Overall, cMIM (i) requires no positive‑pair augmentations, (ii) is less sensitive to batch size, and (iii) improves downstream discrimination while preserving reconstruction across vision and molecular domains. We believe the above additions address the main concerns and further clarify the paper’s contribution.

---

### Meta-Review · Area_Chair_UEQY · 2025-12-26

**Summary:**

This paper propose a contrastive extension of the Mutual Information Machine (MIM) with the goal of unifying discriminative and generative self-supervised learning. It introduces a probabilistic contrastive mechanism that removes the need for positive augmentations and reduces sensitivity to batch size.

The reviewers have various concerns on this submission.  after the rebuttal, the majority of reviewers did not vote the acceptance of the paper. One reviewer lowered the score to 4 even after the rebuttal.  Based on the average score and overall recommendation from the reviewers, I CAN NOT recommend its acceptance.

**Reviewer Concerns:**

The main  concern from my reading of reports are that the current version did not fully  convince the proposed methods are superior to the existing methods. Specifically, reviewers mentioned the lacking of comparisons against VICReg, SimSiam, and BYOL by Reviewer M3RK;  No analysis or comparisons against other augmentation-free contrastive methods (e.g., VICReg and Barlow Twins) by Reviewer Uqjf.  Considering there is a large literature of different contrastive learning approaches, I think the comparisons with the state of the arts are necessary.

**Reviewer Scores:**

no. i do not think that the reviewers will change their score if they had been able to participate fully in the discussion since i think that the comments/concerns raised by them are pretty reasonable and the feedback from the authors provided profound evidence.

---

### Decision · Program_Chairs · 2026-01-26

Reject